# Dynamic acceleration response of a rock slope with a horizontal weak interlayer in shaking table tests

Hanxiang Liu[1,2]*, Tong Qiu[3], Qiang Xu[1,2]

**1** State Key Laboratory of Geohazard Prevention and Geoenvironment Protection, Chengdu University of Technology, Chengdu, China, **2** College of Environment and Civil Engineering, Chengdu University of Technology, Chengdu, China, **3** Department of Civil and Environmental Engineering, Pennsylvania State University, University Park, State College, PA, United States of America

* hxliu_86@163.com

**Data Availability Statement:** All relevant data are within the manuscript and its Supporting Information files.

**Funding:** This research is financially supported by the National Natural Science Foundation of China

## Abstract

The weak interlayer in a rock slope often plays a significant role in seismic rockslides; however, the effect of weak interlayer on the seismic slope response and damage process is still not fully understood. This study presents a series of shaking test tests on two model slopes containing a horizontal weak interlayer with different thicknesses. A recorded Wenchuan earthquake ground motion was scaled to excite the slopes. Measurements from accelerometers embedded at different elevations of slope surface and slope interior were analyzed and compared. The effect of the weak interlayer thickness on the seismic response was highlighted by a comparative analysis of the two slopes in terms of topographic amplification, peak accelerations, and deformation characteristics as the input amplitude increased. It was found that the structure deterioration and nonlinear response of the slopes were manifested as a time lag of the horizontal accelerations in the upper slope relative to the lower slope and a reduction of resonant frequency and Fourier spectral ratio. Test results show that under horizontal acceleration, both slopes exhibited significant topographic amplification in the upper half, and the difference in amplification between slope face and slope interior was more pronounced in Slope B (with a thin weak interlayer) than in Slope A (with a thick weak interlayer). A four-phased dynamic response process of both slopes was observed. Similar deformation characteristics including development of strong response zone and macro-cracks, vertical settlement, horizontal extrusion and collapse in the upper half were observed in both slopes as the input amplitude increased; however, the deformations were more severe in Slope B than in Slope A, suggesting an energy isolation effect of the thick interlayer in Slope A.

## Introduction

The 2008 Wenchuan earthquake triggered more than 60,000 landslides in an elliptical area of approximately 44,000 km$^2$ along the fault rupture zone [1]. There were 112 large destructive

under Grant No. 41702316 and the State Key Laboratory of Geohazard Prevention and Geoenvironment Protection Independent Research Project under Grant No. SKLGP2018Z015. The second author is a collaborator and received no funding from the aforementioned sources but his effort is supported by the US National Science Foundation under award no. CMMI-1453103.

**Competing interests:** The authors have declared that no competing interests exist.

rockslides, each with a surface area exceeding 0.5 km$^2$ [2]. Among the diverse factors, the lithology and slope structure, especially the weak interlayer, played an important role in the initiation of these rockslides. For example, the Daguangbao avalanche in An County, Mianyang City was the largest one with a slide volume of $11.6 \times 10^8$ m$^3$, where the slip surface was located in an interlamination disturbed belt overlain by a 400-m thick of rock stratum [3]. The Donghekou avalanche in Qingchuan County, Guangyuan City was also initiated within a thinbedded weak carbonaceous slate [4]. Much research effort has been directed to exploring the initiation and kinematic mechanisms of these rockslides; however, the effect of weak interlayer remains unclear.

In the past few decades, extensive numerical studies have been carried out to explore the seismic response and deformation/failure behavior of rock slopes, using the finite difference method [5–7], the finite element method [8–10], the discrete element method [11–13], and many others [14, 15]. Numerical studies have advantages in allowing parametric analysis [16]. However, several key problems such as the complex constitution behaviors of geomaterails and boundary conditions still pose great challenges for accurately modeling seismic slope behaviors. In-situ field monitoring has been increasingly utilized to study the seismic behaviors of rock slopes. Del Gaudio and Wasowski carried out the long-term field monitoring of a potential landslide in Italy and observed the directional differences in shaking energy by a factor of 2–3 under several earthquake events [17]. Since then, the directional resonance response and the amplification factor of landslide-prone slopes have been continuously studied with the aid of polarization technique and horizontal-vertical spectral ratio analysis on seismic acceleration waves and ambient noise waves [18–22]. These studies pointed out a combined effect of topographic, lithological, and structural factors on the strong variation in seismic response of different slopes [23–25]. Nonetheless, field monitoring programs are much more costly and time-consuming to obtain sufficient data required for a systematic analysis in comparison with other methods.

Besides numerical simulation and field monitoring, shaking table tests are widely used in studying the seismic behaviors of rock slopes. The majority of test models simulated rock slopes with soil covering [26, 27] or with discontinuous structures, such as jointed rock slopes [28–30], bedding rock slopes [31, 32], counter-bedding rock slopes [10, 33], and horizontally layered rock slopes [34, 35]. These studies generally focused on the dynamic responses of acceleration/displacement and seismic failure, including the topographic/stratigraphic amplification of slope responses and the spatial development of discontinuities and their relations with the seismic forces controlling the failure modes of rock slopes. It was generally observed that the amplitude and angle of incident wave played a more important role in seismic amplification than the duration of shaking did. In contrast, less effort was dedicated to the influence of weak interlayer on the seismic behaviors of rock slopes in shaking table tests [36–39]. Herein, a weak interlayer is defined as an intercalation of a certain thickness between two relatively hard rock layers. Shaking table tests were carried out on a bedding slope and a counter-bedding slope, each of which had six 3-cm-thick weak interlayers [36, 37]. It was found that the acceleration amplification coefficients increased with increasing elevation in both slopes. But when the input earthquake amplitude exceeded 0.3 g, both slopes began to exhibit nonlinear response. In the study [38], a weak silty clay zone was incorporated into a shale rock slope. Because of the influence of the weak zone, the seismic Hilbert energy transferred from the high-frequency components to the low-frequency components and the seismic Hilbert energy in the sliding mass was larger than that in the sliding bed, causing different dynamic responses between the sliding mass and the sliding bed.

The authors conducted a series of shaking table tests on slopes with horizontal bedding planes and different lithology [34, 35], which suggested a significant effect of slope structure

on the seismic responses of a slope. To better understand the complex seismic response and deformation/failure behavior of rock slopes with a weak interlayer during an earthquake, additional shaking table tests were conducted on two model slopes, each with a single horizontal weak interlayer but different thicknesses. The slope above and below the interlayer consisted of the same materials without discontinuities. This simplification excludes the interference of lithology and other types of discontinuities (e.g., bedding planes and joints), which can address the effect of a weak interlayer on seismic slope behaviors. Because the influence of the dip angle of the weak interlayer is not the scope of this study, for simplicity, the dip angle was designed to be0˚ in both model slopes. The recorded horizontal and vertical acceleration responses of the slopes were analyzed in time and frequency domains, based on which the dynamic response process was identified. In the following sections, the shaking table testing program is first described, followed by discussions of the test results and dynamic response process with a focus on the influence of the weak interlayer.

## Test program

### Similitude relations

In the present study, Buckingham's π theory was used to guide the similitude relations between the prototype and model slope. Fifteen parameters are involved in the shaking table tests, including three independent fundamental parameters, namely density, elasticity modulus, and time. Based on the dimension anlysis and π theory [40], the dimensions of these parameters and the π terms are derived and listed in Table 1. The dynamic behavior of rock and soil structure system during an earthquake is governed by basic equations such as constitutive law of the materials, equilibrium equation, and geometric equation [41]; consequently, the scaling factors of related parameters, including displacement ($u$), length ($l$), Young's modulus ($E$), stress ($\sigma$), strain ($\varepsilon$), density ($\rho$), acceleration of gravity (g), and time ($t$) should satisfy relations such as $C_u = C_l C_\varepsilon$, $C_\sigma = C_E C_\varepsilon$, $C_\varepsilon = C_\rho C_g C_l C_E^{-1}$ and $C_t = C_\rho^{0.5} C_l C_E^{-0.5}$, where these terms are defined in Table 2. To satisfy the similitude and account for the large differences in material properties, different values of scaling factors were adopted in this study for the rock material and the weak interlayer material, as shown in Table 2. The scaling factors of the three

**Table 1. Dimensions and dimensionless π terms of key parameters considered in similitude (mass [$M$], length [$L$], time [$T$] are the fundamental dimensions).**

| Number | Parameters | Dimensions | Dimensionless π terms |
|---|---|---|---|
| 1 | Density, $\rho$ | $[\rho] = [M][L]^{-3}$ | controlling parameter |
| 2 | Time, t | $[t] = [T]$ | controlling parameter |
| 3 | Young's modulus, $E$ | $[E] = [M][L]^{-1}[T]^{-2}$ | controlling parameter |
| 4 | Length, $l$ | $[l] = [L]$ | $\pi_l = l/(\rho^{-0.5} E^{0.5} t)$ |
| 5 | Poisson ratio, $\mu$ | $[\mu] = [1]$ | $\pi_\mu = 1$ |
| 6 | Cohesion, $c$ | $[c] = [M][L]^{-1}[T]^{-2}$ | $\pi_c = c/E$ |
| 7 | Friction angle, $\phi$ | $[\phi] = [1]$ | $\pi_\varphi = 1$ |
| 8 | Stress, $\sigma$ | $[\sigma] = [M][L]^{-1}[T]^{-2}$ | $\pi_\sigma = \sigma/E$ |
| 9 | Strain, $\varepsilon$ | $[\varepsilon] = [1]$ | $\pi_\varepsilon = 1$ |
| 10 | Frequency, $f$ | $[f] = [T]^{-1}$ | $\pi_f = f/t^{-1}$ |
| 11 | Displacement, $u$ | $[u] = [L]$ | $\pi_u = u/(\rho^{-0.5} E^{0.5} t)$ |
| 12 | Velocity, $v$ | $[v] = [L][T]^{-1}$ | $\pi_v = v/(\rho^{-0.5} E^{0.5} t)$ |
| 13 | Acceleration, $a$ | $[a] = [L][T]^{-2}$ | $\pi_a = a/(\rho^{-0.5} E^{0.5} t^{-1})$ |
| 14 | Acceleration of gravity, $g$ | $[g] = [L][T]^{-2}$ | $\pi_g = g/(\rho^{-0.5} E^{0.5} t^{-1})$ |
| 15 | Damping ratio, $\lambda$ | $[\lambda] = [1]$ | $\pi_\lambda = 1$ |

Table 2. Scaling factors used in shaking table tests.

| Number | Parameters | Scaling factors (Prototype/Model) | Values (* controlling parameter) | |
|---|---|---|---|---|
| | | | Rock | Weak interlayer |
| 1 | Density, $\rho$ | $C_\rho$ | 1* | 0.67* |
| 2 | Time, t | $C_t$ | 4* | 4* |
| 3 | Elasticity modulus, $E$ | $C_E$ | 32.6* | 10.67* |
| 4 | Length, $l$ | $C_l = C_\rho^{-0.5} C_E^{0.5} C_t C_l$ | 22.9 | 16 |
| 5 | Poisson ratio, $\mu$ | $C_\mu$ | 1 | 1 |
| 6 | Cohesion, $c$ | $C_c = C_E$ | 32.6 | 10.67 |
| 7 | Friction angle, $\phi$ | $C_\phi$ | 1 | 1 |
| 8 | Stress, $\sigma$ | $C_\sigma = C_E C_\varepsilon$ | 22.8 | 10.67 |
| 9 | Strain, $\varepsilon$ | $C_\varepsilon = C_\rho C_g C_l C_E^{-1}$ | 0.7 | 1 |
| 10 | Frequency, $f$ | $C_f = C_t^{-1}$ | 0.25 | 0.25 |
| 11 | Displacement, $u$ | $C_u = C_l C_\varepsilon$ | 16 | 16 |
| 12 | Velocity, $v$ | $C_v = C_u C_t^{-1}$ | 4 | 4 |
| 13 | Acceleration, $a$ | $C_a = C_u C_t^{-2}$ | 1 | 1 |
| 14 | Acceleration of gravity, g | $C_g$ | 1 | 1 |
| 15 | Damping ratio, $\lambda$ | $C_\lambda$ | 1 | 1 |

indepentdent control parameters were determined as $C_\rho = 1$ and $C_t = 4$ for all tests with $C_E = 32.6$ for the rock material and $C_E = 10.67$ for the weak interlayer material.

## Preparation of model slopes

In order to investigate the influence of the weak interlayer on the seismic response of a slope, two model slopes, each with a horizontal weak interlayer, were constructed in a model container, as shown in Figs 1 and 2. In order to minimize the effect of reflected waves from the boundaries, a 20-cm-thick polystyrene sheet was placed on each of the side walls of the container; similar approach has been used in many model tests (e.g., [10, 42, 43]). The thickness of the weak interlayer was 15 cm and 3 cm for Slope A and Slope B, respectively. The soft rock in both model slopes was intended to simulate the rock material of the Zhengjiashan landslide in Pingwu County which was triggered by the 2008 Wenchuan earthquake [2]. The properties of the weak layer were selected based on the properties reported by [44, 45]. Each model slope has a length of 165 cm, a width of 150 cm, a height of 180 cm, and a slope angle of 60° with a smooth slope face. The scaling factor of length $C_l = 22.9$ (see Table 2) suggests that the corresponding prototype slope has a height of 41.2 m.

Barite powder, quartz sands, gypsum, glycerol and water were mixed in weight proportions of 32:56:6:1:7 to produce a low-strength material to simulate the soft rock; while a mixture of quartz sands, Chengdu clay, and water in weight proportions of 20:66:14 was used to simulate the weak interlayer. The above proportions were determined through try and error. The barite powder had a maximum particle size of 0.074 mm and the quartz sands had particle sizes ranging from 0.074 to 0.85 mm. Glycerol was used to slow down the curling of the mixture. Direct shear tests were conducted to obtain the cohesions and internal friction angles of the hardened mixtures while uniaxial compression tests were conducted to obtain the elastic moduli. The material properties are listed in Table 3. Comparing the values of the same parameter between the prototype and model slopes, their ratios (i.e., prototype/model) conformed to the corresponding scaling factors listed in Table 2, with an exception of cohesion, $c$. The cohesions of the rock and weak interlayer were much lower than the values required to satisfy the scaling

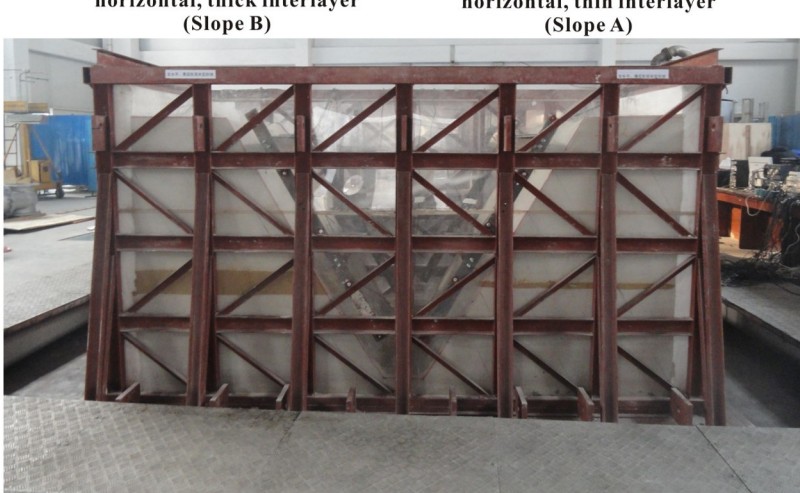

**Fig 1. Two model slopes with a weak interlayer with different thicknesses.**

factors. However, the dynamic behaviors of the model slopes, which are the focus of this present study, are not significantly influenced by the inconformity in cohesion, particularly under small deformations.

## Test setup

The tests were carried out on a large-scale shaking table with six degrees of freedom, including three degrees of translational and three degrees of rotational freedoms. The table is 6 m × 6 m in size and has a maximum load capacity of 60 tons. The maximum acceleration is 1.0 g horizontally and 0.8 g vertically at full load capacity, and 3.0 g horizontally and 2.6 g vertically when unloaded. The maximum displacements in the horizontal and vertical directions are ±150 mm and ±100 mm, respectively, with a loading frequency in the range of 0.1–80 Hz.

As shown in Fig 2, eleven three-component accelerometers with a measuring capacity of ±6.0 g were embedded at different elevations on the surface and inside of each model slope. To observe the topographic effect of acceleration responses, points A1', A2', A3', A4', A5' were located on the outer surface of Slope A and marked as monitoring Line #1. Points A7', A9' and A5' were located in the middle of Slope A and marked as monitoring Line #2. Points A6', A8', A10' and A11' were located in the interior of Slope A and marked as monitoring Line #3. In order to observe the effect of the weak interlayer on the wave propagation, three pairs of points, A2' and A3', A7' and A9', and A8' and A10', were located close to the bottom and top of the weak interlayer in Slope A, respectively. Three differential displacement transducers (LVDT) with a measuring capacity of ±50 mm, named as D1', D2', and D3', were installed at different elevations on the slope surface to measure horizontal displacements of the model slope. The aforementioned monitoring points and lines were similarly instrumented and labelled for Slope B based on their respective positions. To minimize the transverse boundary effect (i.e., perpendicular to the excitation direction), all sensors were placed along the middle profile from toe to top of each slope. An accelerometer (AT1 in Fig 2) was fixed on the bottom plate of the shaking table to monitor the input excitation. Because most LVDTs failed to record displacements during seismic loading, the dynamic displacement responses of both model slopes are not analyzed in this study.

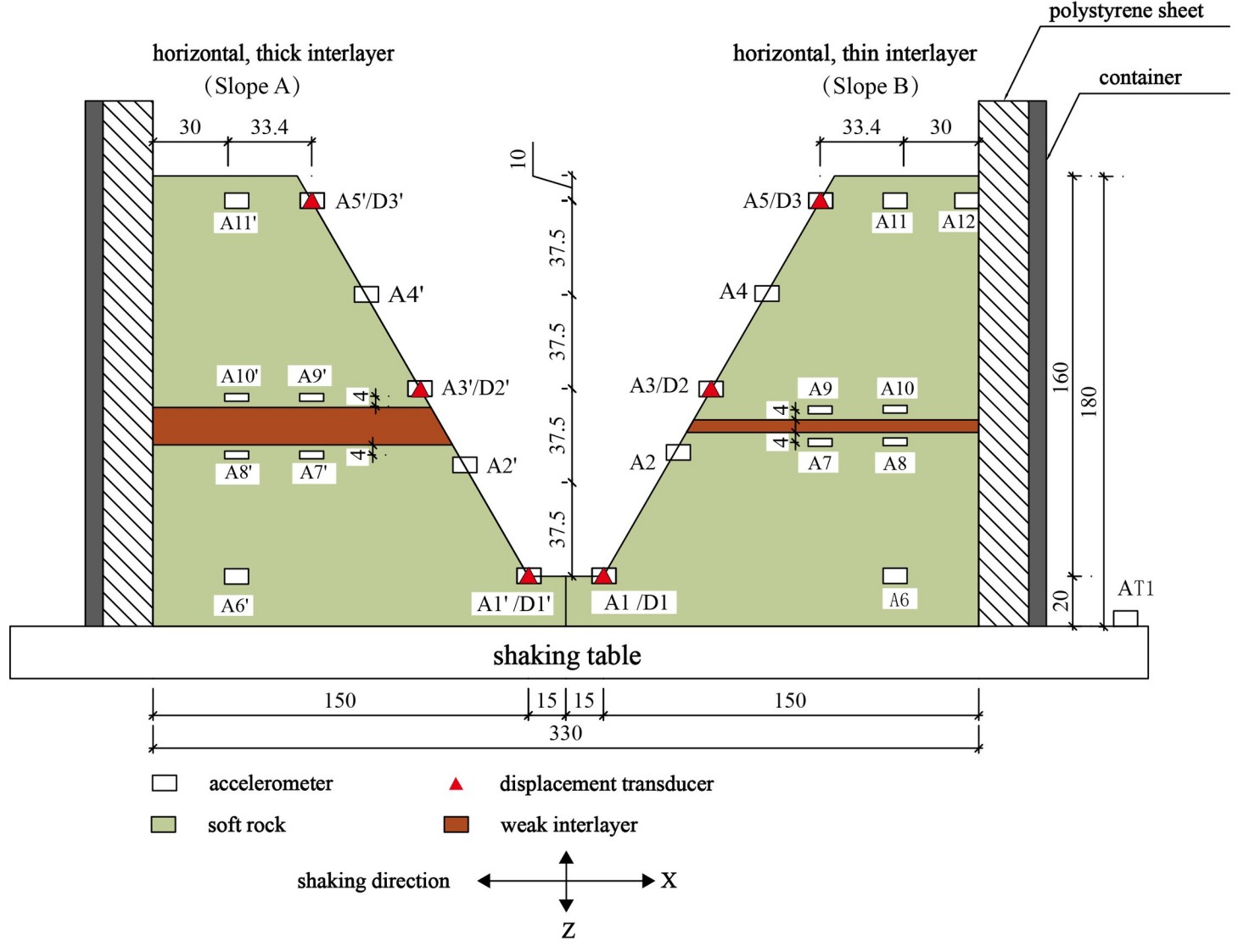

**Fig 2. Schematics of model slopes in shaking table test with sensor arrangement (Unit: Cm).**

### Input excitations

The test process can be divided into two stages, as shown in Table 4. The first stage is designed to mainly study the acceleration responses of the model slopes under different amplitudes of excitations. It is widely known that loading history has a significant impact on the dynamic response of the model slopes. As a widely-adopted mitigation to reduce this impact, waves

**Table 3. Physical and mechanical properties of prototype slopes and model slopes.**

| Lithology | | Density $\rho$ ($10^3$kg/m$^3$) | Cohesion $c$ (kPa) | Frictional angle $\phi$ (°) | Elastic modulus $E$ (MPa) | Poisson's ratio $\mu$ |
|---|---|---|---|---|---|---|
| Rock | Prototype | 2.48 | 520 | 33.8 | 1900.0 | 0.31 |
| | Model | 2.40 | 37.1 | 34.9 | 50.2 | 0.30 |
| Weak interlayer | Prototype | 1.60 | 30.0 | 35.0 | 41.0 | 0.35 |
| | Model | 2.32 | 10.0 | 27.3 | 4.8 | 0.35 |

**Table 4. Loading scheme of shaking table tests.**

| loading step | Wave type | Amplitude (g) | Excitation direction | Duration (s) | Dominant frequency (Hz) |
|---|---|---|---|---|---|
| First loading stage | | | | | |
| 1 | WL wave | 0.1 | Z | 27 | 32.4 |
| 2 | WL wave | 0.1 | X | 27 | 9.6 |
| 3 | WL wave | 0.18 | Z | 27 | 32.4 |
| 4 | WL wave | 0.2 | X | 27 | 9.6 |
| 5 | WL wave | 0.24 | Z | 27 | 32.4 |
| 6 | WL wave | 0.3 | X | 27 | 9.6 |
| 7 | WL wave | 0.32 | Z | 27 | 32.4 |
| 8 | WL wave | 0.4 | X | 27 | 9.6 |
| 9 | WL wave | 0.38 | Z | 27 | 32.4 |
| 10 | WL wave | 0.5 | X | 27 | 9.6 |
| 11 | WL wave | 0.6 | Z | 27 | 32.4 |
| 12 | WL wave | 0.8 | X | 27 | 9.6 |
| 13 | WL wave | 0.75 | Z | 27 | 32.4 |
| 14 | WL wave | 1.0 | X | 27 | 9.6 |
| Second loading stage | | | | | |
| 15 | Sine wave | 0.55 | X | 27 | 10 |
| 16 | Sine wave | 0.6 | X | 27 | 10 |
| 17 | Sine wave | 0.7 | X | 27 | 10 |
| 18 | Sine wave | 0.8 | X | 27 | 10 |

were applied to the model slopes with a gradually increasing amplitude [29–39]. The loading sequence in Table 4 allowed an insight into the evolution of the slope response from linear to nonlinear, assuming that the effect of minor deformation of model slope can be neglected under small-amplitude excitations. In addition, the cumulative damage caused by continuous loading could partially simulate pre-existing fractures/fissures in a prototype slope before a seismic event.

The input accelerations were scaled from the recorded accelerations at the Wolong seismic station (WL wave) during the 2008 Wenchuan earthquake, which was about 23 km southwest of the epicenter. The altitude on the station ground is 919 m above sea level whereas the altitude on the top of the mountain nearby is 3187 m. The overlying soils at the station are mainly composed of Quaternary alluvial and diluvial gravels and pebbles. The recorded accelerations had a dominant frequency of 2.4 Hz and 8.1 Hz in the horizontal and vertical direction, respectively, which were scaled to have dominant frequency of 9.6 Hz and 32.4 Hz, respectively (see Table 4). The scaled amplitudes of the horizontal and vertical accelerations were gradually increased from 0.1 g to 1.0 g and 0.1 g to 0.75 g, respectively. The difference in peak accelerations between the horizontal and vertical directions was due to the capacity of the shaking table under high levels of excitations. Fig 3 shows the input horizontal and vertical accelerations under 0.1 g amplitude recorded by AT1 (see Fig 2) and their Fourier amplitude spectrums.

The second stage is to carry out destructive test and several sine waves with a frequency of 10 Hz in the X direction were used as the input motion until the slopes demonstrated structural damage. The initial loading amplitude of the sine wave was 0.55 g, an excitation level known to cause large deformation according to previous studies [35]. Each step lasted for about 27 s which was scaled at 1:4 according to the mainshock duration (108 s) of the Wenchuan earthquake at the Wolong seismic station. Between two consecutive loading steps, there was a short break to examine if the model slopes failed or had any large deformation.

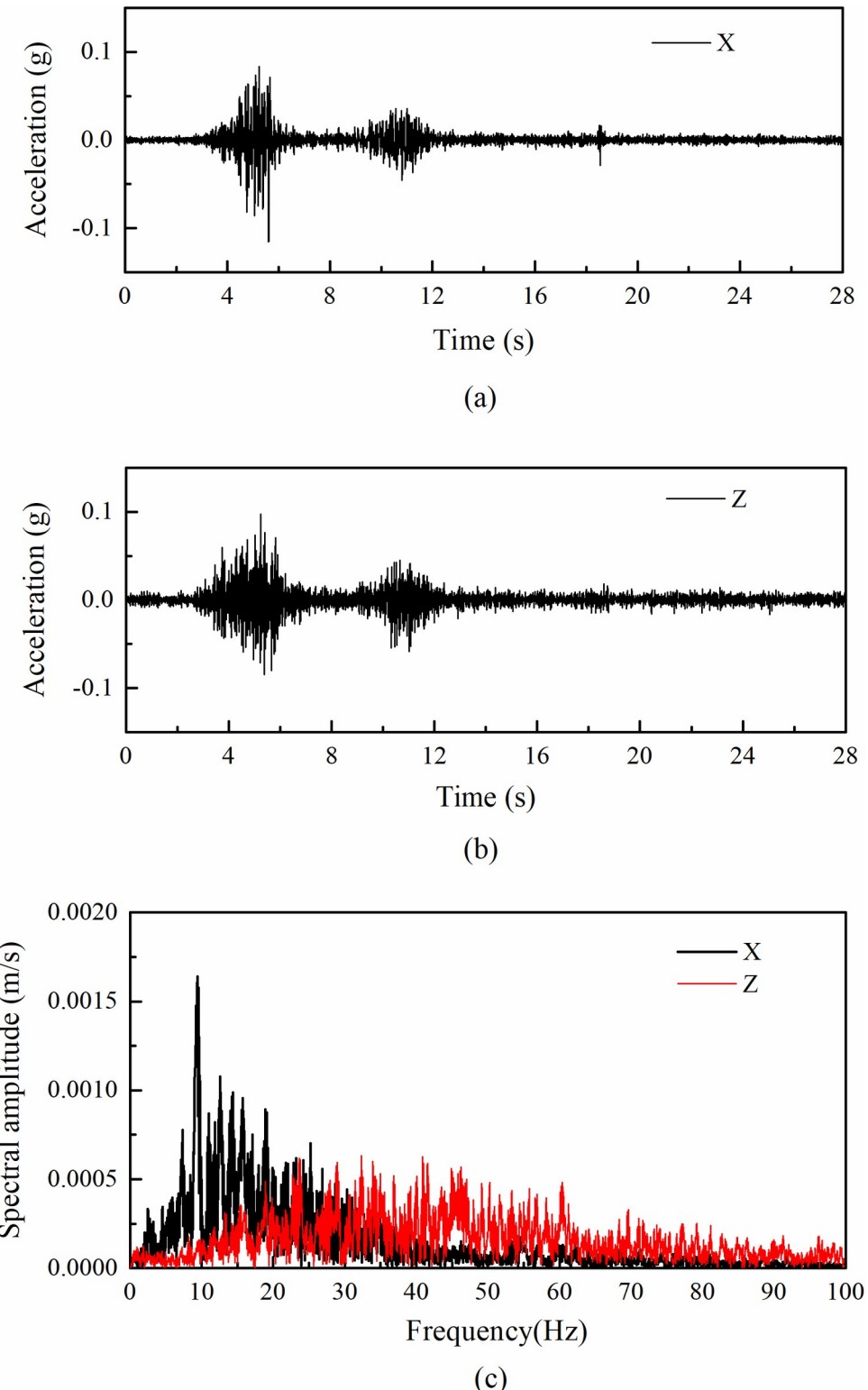

**Fig 3.** Input ground motion in shaking table tests for the case of 0.1 g input amplitude: (a) horizontal acceleration time histories; (b) vertical acceleration time histories; (c) corresponding Fourier spectrum.

## Acceleration responses of slopes

### Horizontal acceleration response

Fig 4 shows the horizontal acceleration responses of several monitoring points under 0.1 g and 0.8 g levels of horizontal shaking in both time domain and frequency domain for both slopes.

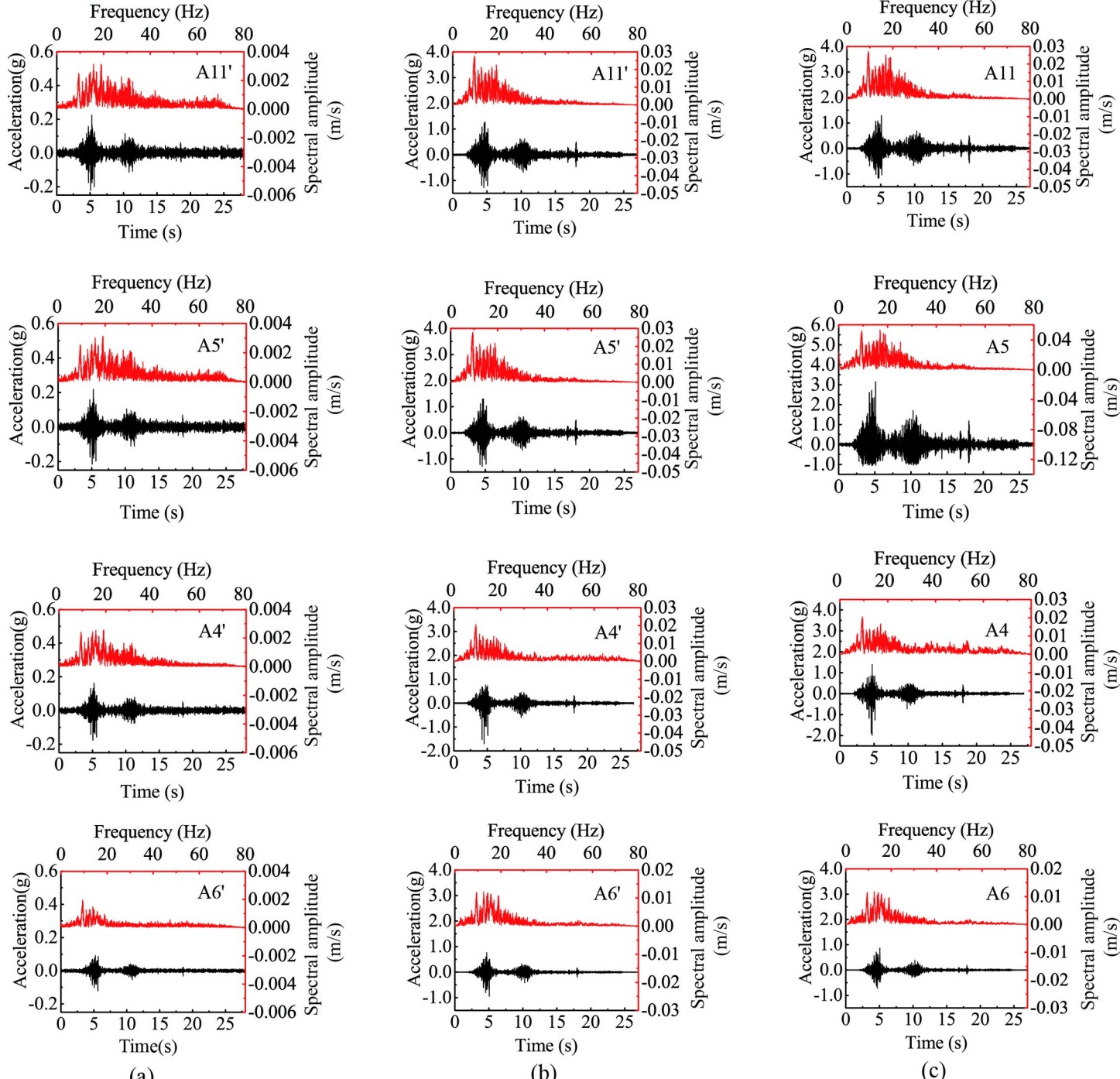

**Fig 4.** Horizontal acceleration responses at different monitoring points of two model slopes under 0.1 g and 0.8 g levels of horizontal shaking: (a) input amplitude = 0.1 g, Slope A, (b) input amplitude = 0.8 g, Slope A, (c) input amplitude = 0.8 g, Slope B.

Points A6' and A6 represent the responses of the lower slope; Points A4', A5', A4, and A5 represent the responses of the upper slope face; and Points A11' and A11 represent the responses beneath the top surface. Fig 4 shows that the acceleration time histories of these points were generally consistent with the horizontal input motion with all showing two distinctive strong shocks. The responses of the upper slope (i.e., Points A4', A4, A5', A5, A11', and A11) were stronger than those in the lower slope (i.e., Points A6' and A6), indicating a topographic amplification effect. Fig 4 also shows that the Fourier spectral amplitude generally increased along elevation due to the topographic amplification effect. As the input amplitude increased from 0.1 g to 0.8 g, the peak spectral amplitudes of all monitoring points significantly increased (e.g., comparing Fig 4(A) and 4(B)). The distribution of spectral energy depends on the combination of elevation and input amplitude. Comparing to those points in the lower slope (e.g., Point A6' in Fig 4(A)), the points in the upper slope(e.g., Points A4', A5', and A11' in Fig 4(A)) had a higher spectral energy in high frequencies close to 20 Hz; whereas comparing with the excitation at 0.1 g level (e.g., Point A5' in Fig 4(A)), the 0.8-g excitation produced stronger responses in low frequencies close to 10 Hz (e.g., Point A5' in Fig 4(B)).

The effect of strong seismic shaking on the slope response is manifested into two aspects: the time lag of strong shocks in different parts of the slope and the different responses between outward and inward movements at a given monitoring point. In the subsequent analysis, horizontal acceleration out of slope (i.e., outward direction) is positive and towards slope (i.e., inward direction) is negative. Fig 5 shows the enlarged time history during the time frame from 4.5 s to 4.9 s in Fig 4. Fig 5(A) shows that, under 0.1 g level of shaking, Points A4', A5', A6' and A11' experienced synchronous horizontal acceleration with a minor difference between the negative and positive accelerations, indicating a linear response with no permanent deformation occurring in Slope A. Under 0.8 g level of shaking, however, Points A4', A5', and A11' exhibited an evident time lag relative to Point A6', suggesting a nonlinear response of the model slope in the upper slope. Moreover, the large difference between the negative and positive accelerations at Points A4', A5'and A11' depicted an imbalance between inward and outward movements under 0.8 g level of shaking, suggesting a permanent structural deterioration of the slope (macro cracking was not observed). The structure deterioration caused a reduction in shear wave velocity (stiffness), hence, the time lag in response. Similar phenomena were also observed in Slope B (Fig 5(B)).

To quantify the topographic amplification effect, an amplification factor (R-PHA) is defined as the ratio of the peak horizontal acceleration (PHA) measured at a monitoring point to that measured at AT1; hence, R-PHA>1.0 indicates amplification, R-PHA = 1.0 indicates non-amplification, and R-PHA<1.0 indicates de-amplification. A normalized elevation ($h/H$) is defined as the ratio of height $h$ (measured from the toe of the model slope) of a monitoring point to the total height ($H$) of the model slope. Fig 6 shows the variation of R-PHA with elevation for Monitoring Lines#1 (slope face) and #3 (slope interior) under 0.1 g and 0.8 g levels of shaking. The accelerometers at Points A1 and A9 of Slope B failed to record the horizontal accelerations and were excluded in the following analysis. Fig 6 shows that all R-PHAs were greater than 1.0 and generally increased with $h/H$. In addition, the R-PHA increased faster with elevation in the upper half of the model slope than in the lower half, indicating a more distinctive topographic amplification effect in the upper half which is consistent with observations in many model tests (e.g., [26, 27, 37, 38, 46]). Another observation is the effect of input amplitude on the topographic amplification. Under 0.8 g level of shaking (see Fig 6(B)), the amplification on the slope face became much more significant than that of slope interior, particularly for Slope B. Regardless of input acceleration, Fig 6 shows that the difference in amplification between slope face and slope interior is more pronounced in Slope B than Slope A, highlighting the effect of the thickness of the weak interlayer as less wave energy was able to

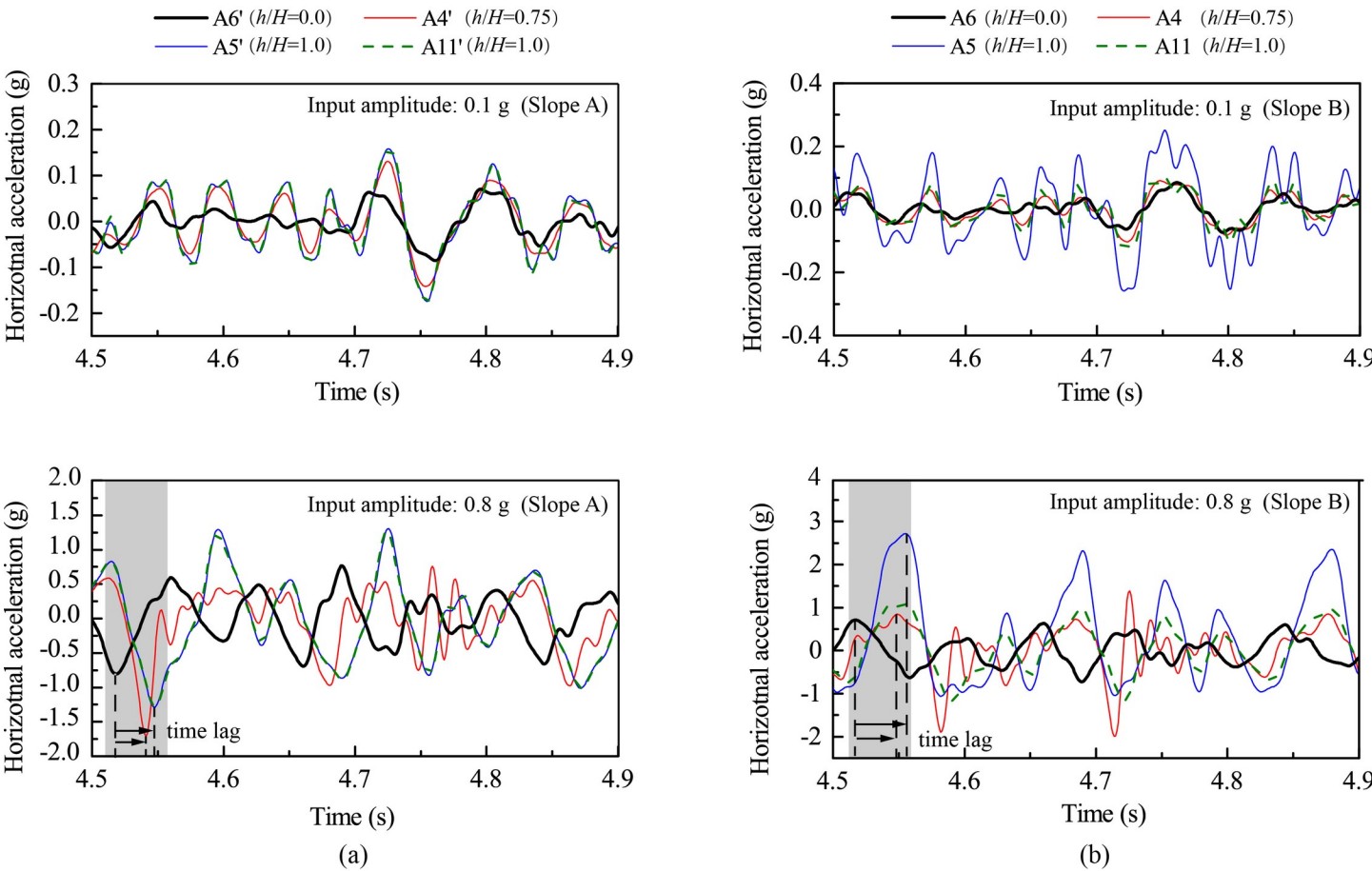

**Fig 5.** Horizontal acceleration responses at selected monitoring points under 0.1 g and 0.8 g levels of horizontal shaking: (a) Slope A; (b) Slope B.

pass through this thicker interlayer which caused a more significant reduction of R-PHA above the interlayer in Slope A.

Fig 7 shows the relationships between PHA, R-PHA, and input amplitude for all monitoring points. For Slope A (see Fig 7(A)), the PHAs exhibited a steady increase with input amplitude from 0.1 g to 1.0 g except that those at Point A4' increased sharply when the input amplitude exceeded 0.5 g. For Slope A, the relationships of R-PHA vs. input amplitude exhibits three patterns, corresponding to three different regions of the slope. The first region is the top slope containing Points A5' and A11' and their surroundings. Strong acceleration response occurred in this region, and the R-PHA experienced a gradual decrease when the input amplitude increased up to 0.8 g, indicating a nonlinear response. Moderate permanent slope deformation was observed in this region under high-amplitude excitations. The second region contains Point A4' and its surroundings. When the input amplitude was smaller than 0.3 g (inclusive), the slope response was linear and the R-PHA exhibited a small increase with input amplitude; when the input amplitude exceeded 0.5 g, the slope response became nonlinear and the R-PHA exhibited a sharp increase with input amplitude; and for input amplitude between 0.3 g and 0.5 g, the slope response was in a transition from linear to nonlinear responses and the R-PHA exhibited a small decrease. The third region contains the weak interlayer, its surrounding region, and the underlying slope, including Points A1', A2', A3', A6', A7', A8', A9' and A10'; in this region, the R-PHAs were very low with a less than 50% of increment relative

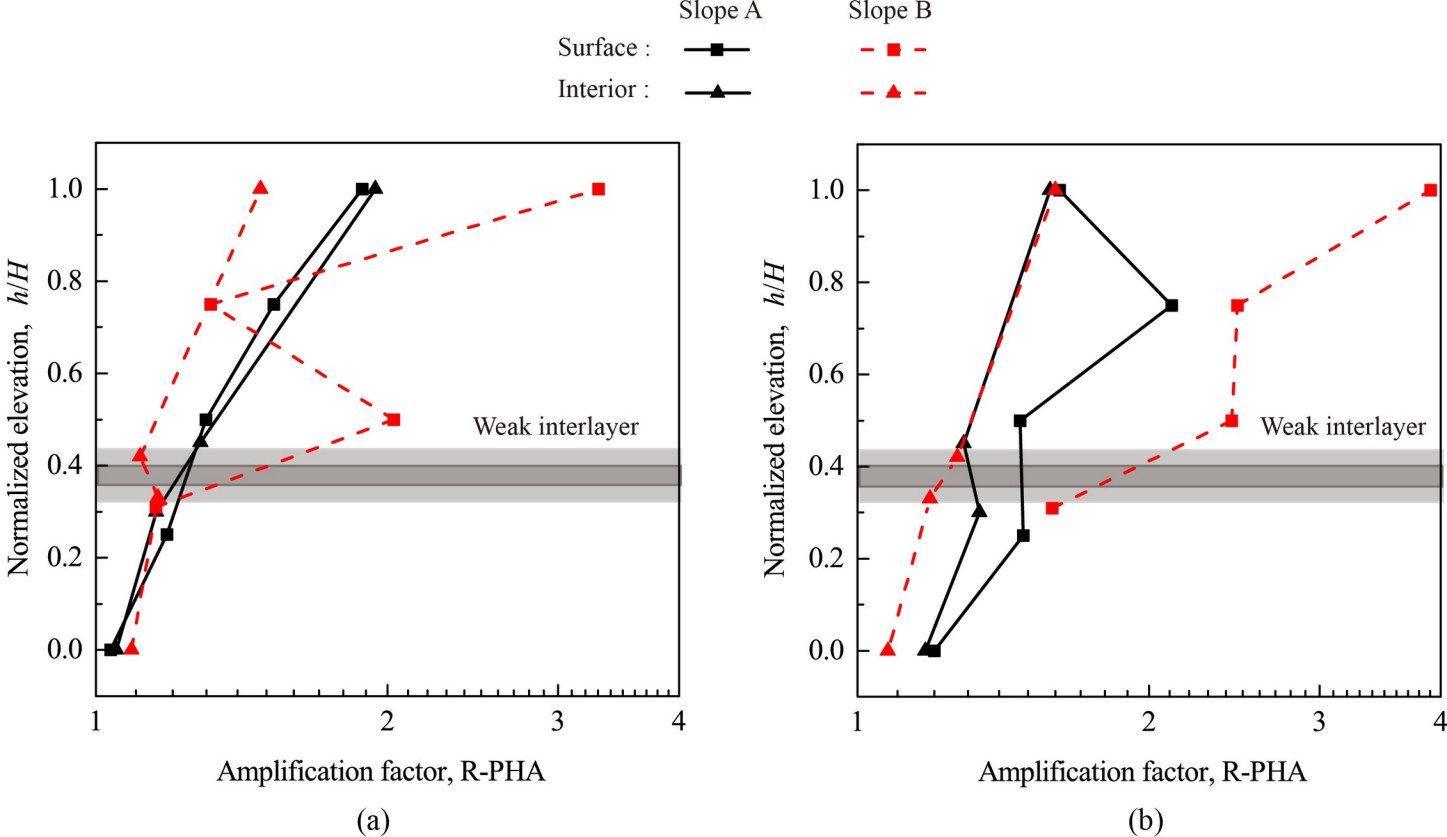

**Fig 6.** Topographic amplification of peak horizontal acceleration (R-PHA) under 0.1 g and 0.8 g levels of horizontal shaking: (a) 0.1 g; (b) 0.8 g.

to the input amplitude, indicating a very small amplification. As input magnitude increased, the slope response in this region remained linear with no permanent deformation.

Slope B exhibited similar responses as Slope A with a few notable exceptions. First, the crest of Slope B (i.e., Point A5), instead of Point A4' for the case of Slope A, exhibited the largest response under all input amplitude levels. Under strong horizontal excitations, Points A4 and A5 (and their vicinity) experienced more significant amplifications, which led to an earlier occurrence of cracking in the upper slope in Slope B than in Slope A (to be discussed later).

## Vertical acceleration response

The vertical response of the model slopes is similarly analyzed. Fig 8 shows the vertical acceleration responses of two monitoring points under 0.1 g and 0.6 g levels of vertical shaking in both time domain and frequency domain for Slopes A and B. Points A1' and A1 represent the responses of the lower slope while Points A5' and A5 represent the responses of the upper slope. Similar to the observations in Fig 4, the vertical acceleration time histories were generally consistent with the vertical input motion with all showing two distinctive strong shocks. Under the two levels of shakings, topographic amplification effect is manifested in both time histories and Fourier amplitude spectrums with larger acceleration amplitudes in the upper slope (i.e., Points A5' and A5) than in the lower slope (i.e., Points A1' and A1). Different from the horizontal spectrums in Fig 4, the spectral energy of vertical acceleration was mainly concentrated between 20 Hz and 40 Hz.

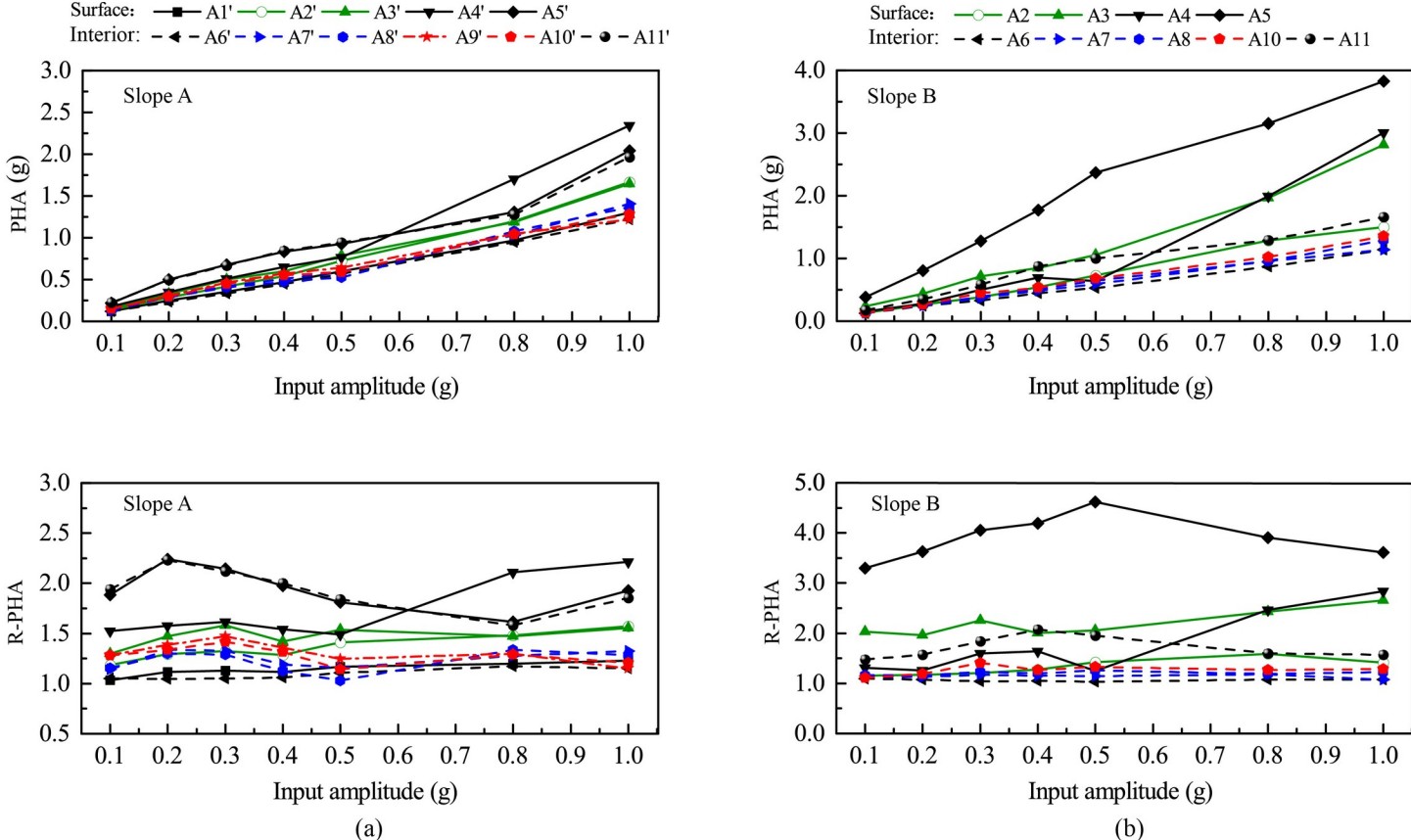

**Fig 7.** Peak horizontal acceleration (PHA) and corresponding amplification factor (R-PHA) of all monitoring points vs. amplitude of horizontal input acceleration: (a) Slope A; (b) Slope B.

In the subsequent analysis, downward vertical acceleration is positive and vice versa. Fig 9 shows the enlarged time history responses during the time frame from 5.1 s to 5.4 s from Fig 8. Fig 9(A) shows that, under 0.1 g level of vertical shaking, Points A1', A4', A5' and A11' of Slope A experienced synchronous vertical acceleration with a minor difference between the negative and positive accelerations. Under 0.6 g level of vertical shaking, however, Points A4', A5', and A11' exhibited a small time lag relative to Point A1'. In contrast, Fig 9(B) shows that, regardless of input amplitude, synchronous vertical acceleration existed at Points A1, A4, A5 and A11 of Slope B, indicating a linear response. Under 0.6 g level of shaking, the positive accelerations (i.e., downward) of both slopes were larger than the negative ones (i.e., upward), especially at Points A5 and A5', suggesting that a downward accumulative vertical deformation.

Fig 10 shows the variation of peak vertical acceleration (PVA) and its amplification factor (R-PVA) with elevation for Monitoring Lines #1 (slope face) and #3 (slope interior) under 0.1 g and 0.6 g levels of shakings. Fig 11 shows the relationships between PVA and R-PVA and input amplitude for all monitoring points. Similar to Figs 6 and 7, a general topographic amplification of vertical acceleration was observed in both slopes, and the amplification pattern was influenced by the input amplitude. A few notable differences in slope responses under vertical and horizontal accelerations can be identified. First, a large difference in amplification between slope surface and slope interior for Slope A occurred in the lower half of the slope under vertical acceleration (see Fig 10) instead of in the upper half of the slope under horizontal acceleration (see Fig 6). Second, Point A4' in Slope A and Point A4 in Slope B didn't experience an

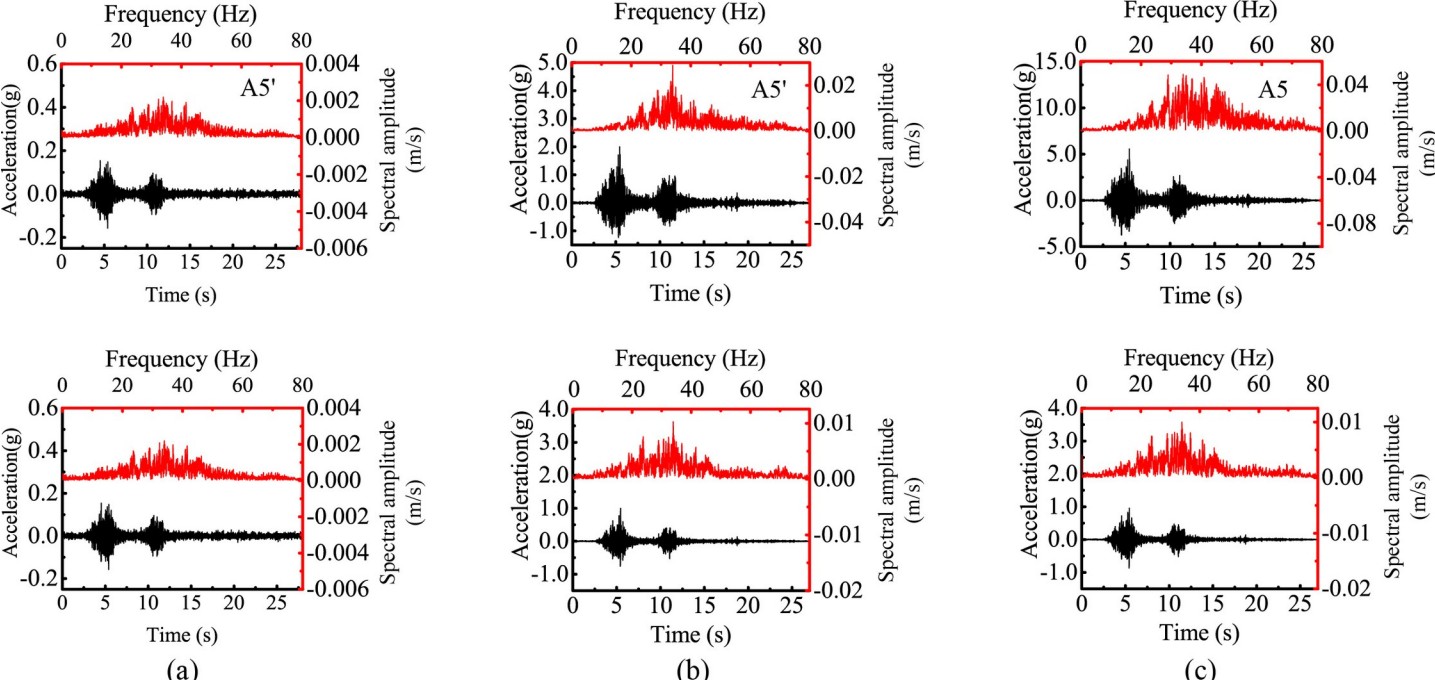

**Fig 8.** Vertical acceleration responses at two monitoring points of two model slopes under 0.1 g and 0.6 g levels of vertical shaking: (a) input amplitude = 0.1 g, slope A; (b) input amplitude = 0.6 g, slope A; (c) input amplitude = 0.6 g, slope B.

abrupt increase in PVA and R-PVA with input amplitude under vertical acceleration (see Fig 11) as they did under horizontal acceleration (see Fig 7). Last, the largest amplification of PVA occurred at the crest of both slopes (e.g., Points A5' and A5) under vertical acceleration (see Fig 11) whereas the largest amplification of PHA occurred at Point A4' in Slope A under horizontal acceleration. Point A5 in Slope B experienced much larger PVA and R-PVA than the rest of the monitoring points (see Fig 11(B)) and the cause is unknown.

Fig 11(A) shows that for Slope A: (1) the monitoring points on slope surface exhibited a similar trend, and (2) the monitoring points on slope interior exhibited a similar but different trend than those on slope surface. On the other hand, Fig 11(B) shows that, for Slope B, all monitoring points exhibited a similar trend. These observations are corroborated by Fig 9 that slope surface and slope interior experienced synchronous vertical acceleration for Slope B whereas there was a minor time lag between slope surface and slope interior for Slope A.

## Fourier spectral ratios

In this section, the structure deterioration of the model slopes in the first loading stage is discussed using the recorded information in the frequency domain. Fourier amplitude spectral ratio (FASR) of accelerations were used in previous studies to identify the resonant frequency and study the nonlinearity characteristics based on ground motion recordings [20, 25, 47]. In this study, FASR is calculated as the Fourier amplitude in the model slope divided by that on Point AT1 (see Fig 2). Fig 12 shows the contours of vertical FASR (V-FASR) along different elevations under 0.1 g, 0.32 g, and 0.75 g levels of vertical shaking. A resonant-frequency band can be identified from Fig 12 to describe the concentrated frequencies at which significant peaks in V-FASR occurred at different elevations. Fig 12 shows that, as the input amplitude increased, the resonant frequency decreased from a band of 35–50 Hz to 20–30 Hz for Slope A

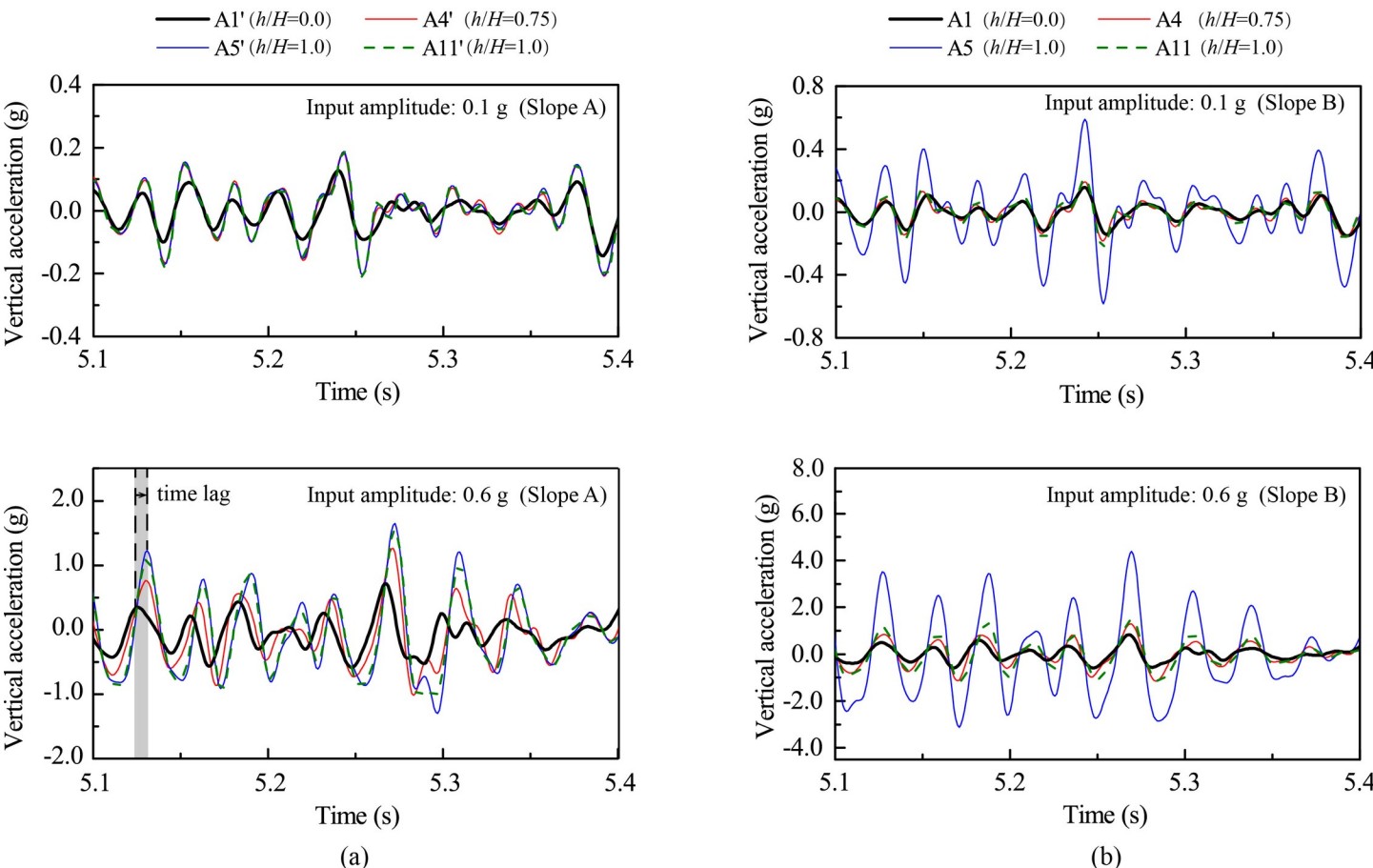

**Fig 9.** Vertical acceleration responses at selected monitoring points under 0.1 g and 0.6 g levels of vertical shaking: (a) Slope A; (b) Slope B.

(see Fig 12(A)) and from a band of 50–60 Hz to 20–30 Hz for Slope B (see Fig 12(B)), suggesting the structure deterioration (i.e., stiffness reduction) of both slopes under increased vertical shaking, which is corroborated by Fig 9. Fig 13 shows the corresponding contour plots under horizontal shaking. A comparison between Figs 12 and 13 shows the slope response under horizontal shaking is much more complex than under vertical shaking, likely due to higher modes of resonance; however, a reduction in resonant frequency at the top of slope is evident for Slope A as the input amplitude increased from 0.1 g to 0.4 g.

## Discussions

In this study, various parameters in time and frequency domains including peak acceleration (i.e., PHA and PVA), acceleration amplification factor (i.e., R-PHA and R-PVA), Fourier amplitude spectral ratio (i.e., H-FASR and V-FASR) and resonant frequency are used to analyze the slope response process. In the first loading stage, the spatial variations of these parameters within each model slope and their evolution with increasing input amplitude suggested that the model slopes experienced a three-phase response process from linear to nonlinear behavior. In order to observe macro failure of the slopes qualitatively, destructive tests were conducted in the second loading stage (see Table 4). Different levels of horizontal sine waves with a frequency of 10 Hz were used to excite the model slopes. As the input amplitude increased, large deformations (e.g., settlement, extrusion, collapse) occurred and were

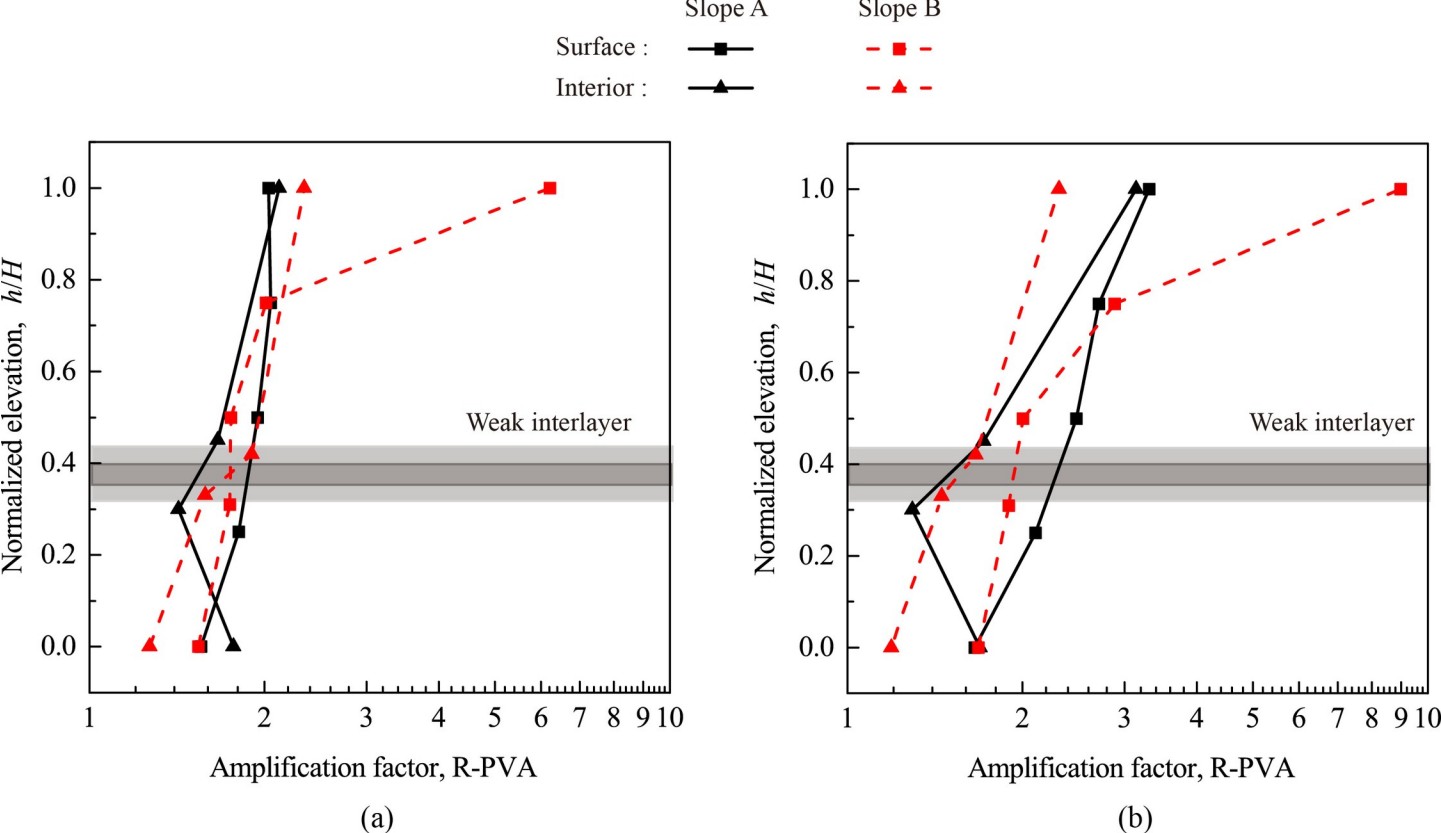

**Fig 10.** Topographic amplification of peak vertical acceleration (R-PVA) under 0.1 g and 0.6 g levels of vertical shaking: (a) 0.1 g; (b) 0.6 g.

recorded. A conceptual model was proposed to describe the dynamic response process for both model slopes and is shown in Fig 14. In the subsequent analysis, the dynamic response process is described on the basis of horizontal input amplitude since the slope deformation was largely induced by horizontal shaking.

Phase I is the linear response phase, corresponding to input amplitudes lower than 0.3 g. This same cutoff value was reported in the literature [26, 36, 46, 48]. In this phase, both the horizontal and vertical accelerations (e.g., PHA, R-PHA, PVA, R-PVA) at most monitoring points of the slopes generally increased as the input amplitude increased. The maximum acceleration was observed at the crest of each slope, causing a strong response zone (i.e., zone①; in Fig 14). Minor change of resonant frequency occurred in both slopes when the input amplitude increased from 0.1 g to 0.3g, especially under horizontal shaking (see S1 Fig). No macro deformation occurred in any parts of the slopes.

Phase II marks the transition phase from linear to nonlinear responses, corresponding to input amplitudes between 0.3 g and 0.5 g. The accelerations at different points of each slope showed inconsistent trends as the input amplitude increased, especially under horizontal shaking. The structure deterioration was manifested by a reduction of horizontal and vertical resonant frequency of both slopes (see Figs 12 and 13). No macro deformation was observed in this phase, but the strong response zone expanded due to the increased shaking level.

Phase III is the nonlinear response phase, corresponding to input amplitudes larger than 0.5 g. In this phase, macro cracks parallel to the slope developed at the top of the slope and slope face. The cracking was more severe in Slope B than in Slope A, likely due to more seismic

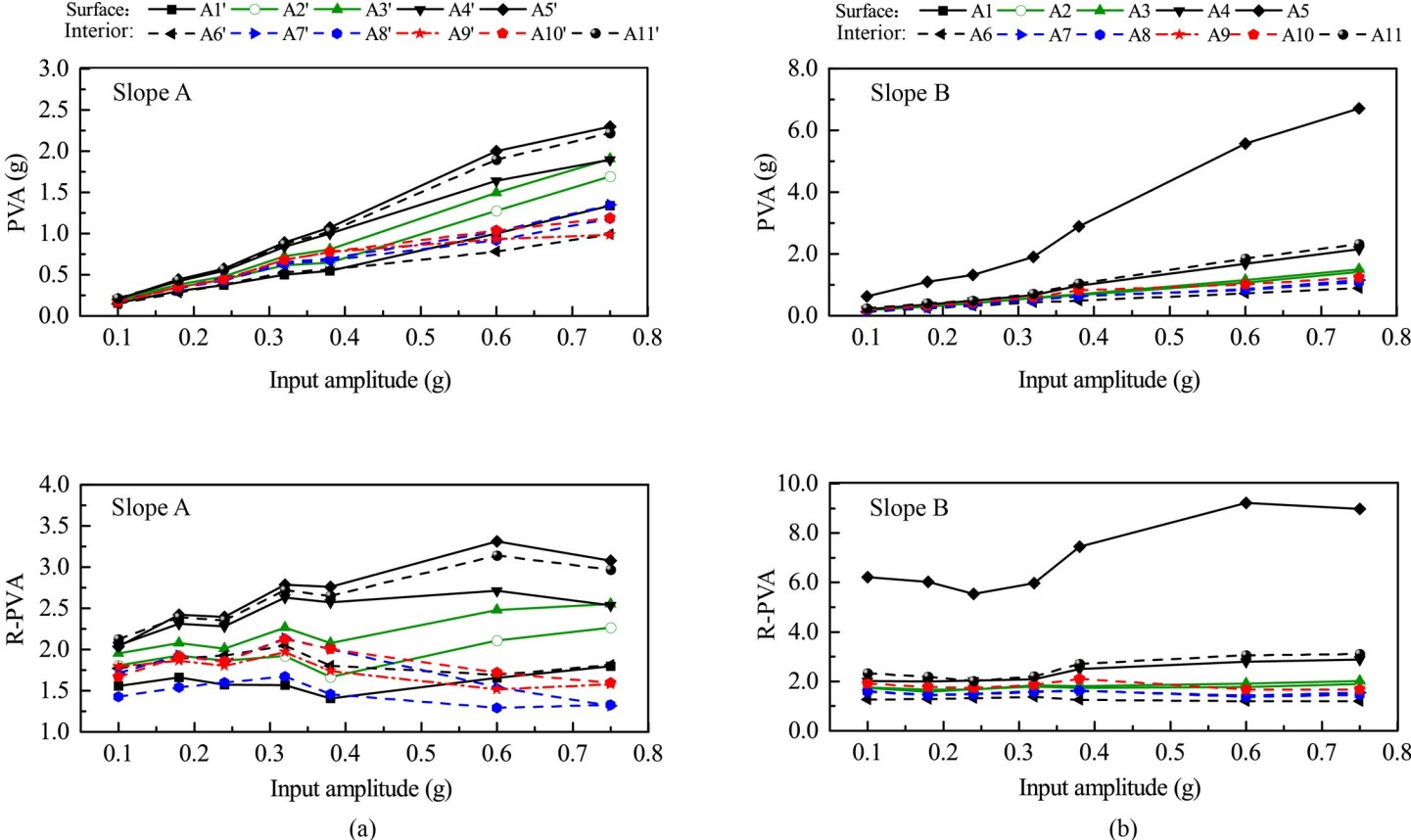

**Fig 11.** Peak vertical acceleration (PVA) and corresponding amplification factor (R-PVA) of all monitoring points vs. amplitude of vertical input acceleration: (a) Slope A; (b) Slope B.

energy transmitted to the upper slope from the thinner weak layer in Slope B. The progressive development of micro and macro cracks remarkably changed the dynamic properties (i.e., modulus, damping, strength) of the slope materials in the strong response zone, causing an inelastic and nonlinear response. In contrast, the lower part of both slopes, to a larger extent, remained elastic and linear due to the relatively small magnitude of shaking (see Fig 7). Structure deterioration and nonlinear response in the upper part of both slopes were manifested by the time lag relative to the lower part and an imbalance between inward and outward movements of horizontal accelerations in time domain (see Fig 5) and the reduction of resonant frequency and spectral ratio in frequency domain (see Figs 12 and 13).

Phase IV presents the final failure of both slopes after the second loading stage. Permanent deformation continued to accumulate in the strong response zone, especially in the surficial region containing Points A4 and A5 in Slope A and Points A4' and A5' in Slope B. For Slope A, a shallow collapse occurred just beneath the crest, severe cracking deformation occurred on the slope surface, and a small horizontal extrusion deformation occurred around the weak interlayer (see Figs 15(A) and 15(C) and S2 Fig). More severe collapse and cracking but a smaller extrusion deformation were observed in Slope B (see Figs 15(B) and 15(D); S2 Fig). It should be noted that the exact boundaries between response zones with different damage degrees (i.e., dash lines in Fig 15) require adequate deformation monitoring data with sufficient spatial resolution, which is not included in the present study. The completely quantitative analysis on the seismic slope responses and failure modes and mechanisms relies on multi-

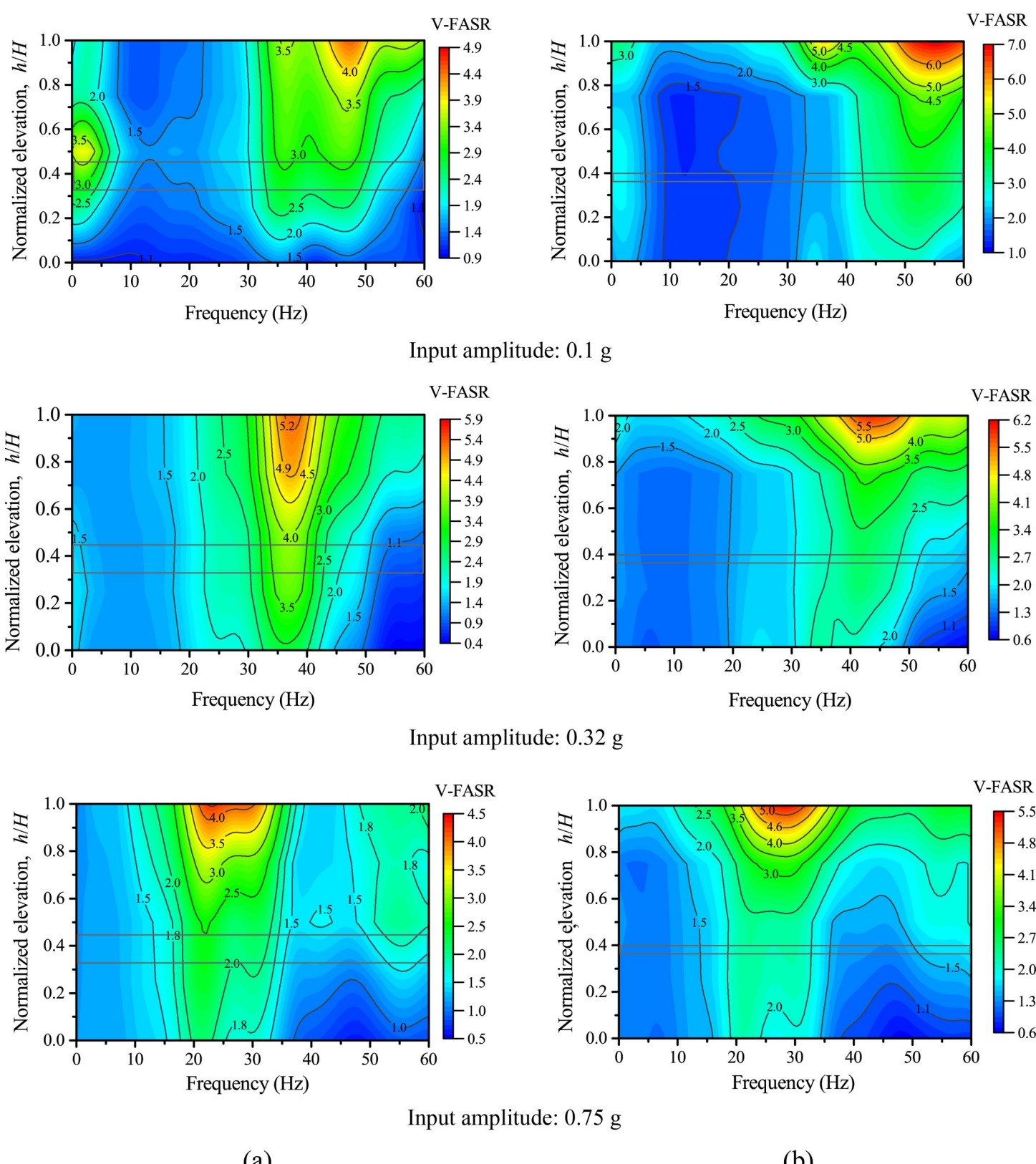

Input amplitude: 0.1 g

Input amplitude: 0.32 g

Input amplitude: 0.75 g

(a)                                                      (b)

**Fig 12.** Fourier amplitude spectral ratios of vertical accelerations (V-FASR) under different levels of vertical shakings: (a) Slope A; (b) Slope B.

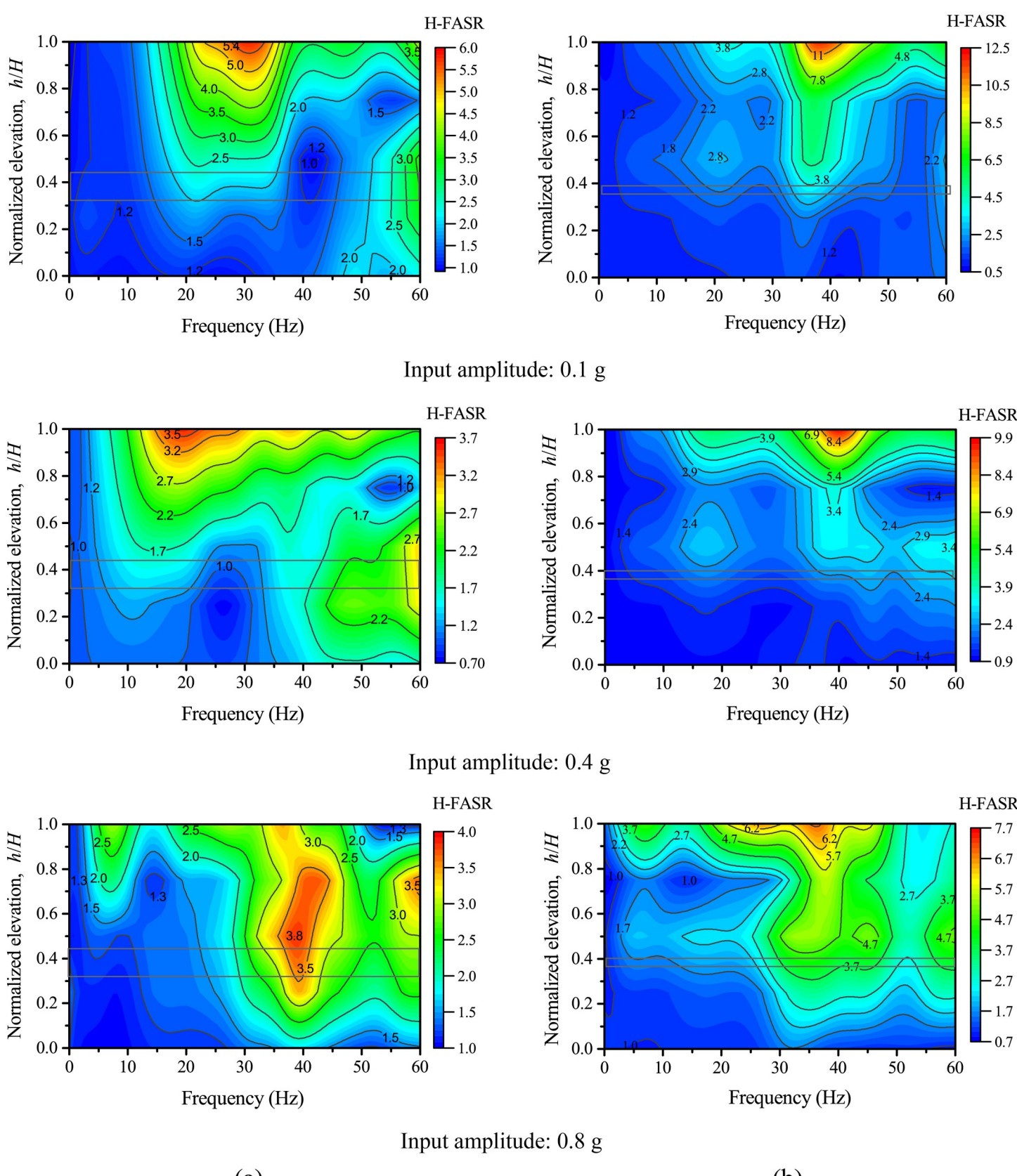

**Fig 13.** Fourier amplitude spectral ratios of horizontal accelerations (H-FASR) under different levels of horizontal shaking: (a) Slope A; (b) Slope B.

| Phases | Response features | Conceptualized dynamic damage process |
|---|---|---|
| Phase I | Linear response<br>No macro deformation | Slope A    Slope B |
| Phase II | Transition response<br>No macro deformation | Slope A    Slope B |
| Phase III | Nolinear response<br>Visible macro deformation | cracks<br>Slope A    Slope B |
| Phase IV | Failure | cracks    settlement    cracks<br>extrusion    extrusion<br>deposite<br>Slope A    Slope B |

—— Pre-test topography    —— Post-test topography    ⊡① Strong response/deformation zone (speculated)

**Fig 14. Conceptualized dynamic response process of two model slopes.** Dash line indicates the speculated strong response/deformation zone but not slip surface.

source data from macro- and micro- numerical calculation, physical model testing, field monitoring, and theoretical analysis, which will be considered in the future work.

## Conclusions

This paper presents an experimental investigation on the effect of a horizontal weak interlayer on the seismic behavior of a rock slope. Two model slopes were constructed and tested simultaneously on a shaking table. These slopes were identical except that the horizontal weak interlayer had a different thickness. The following conclusions specific to the test conditions are reached:

(1) Under horizontal acceleration, both slopes exhibited significant topographic amplification in the upper half, and the difference in amplification between slope face and slope interior was more pronounced in Slope B (with a thin interlayer) than in Slope A (with a thick

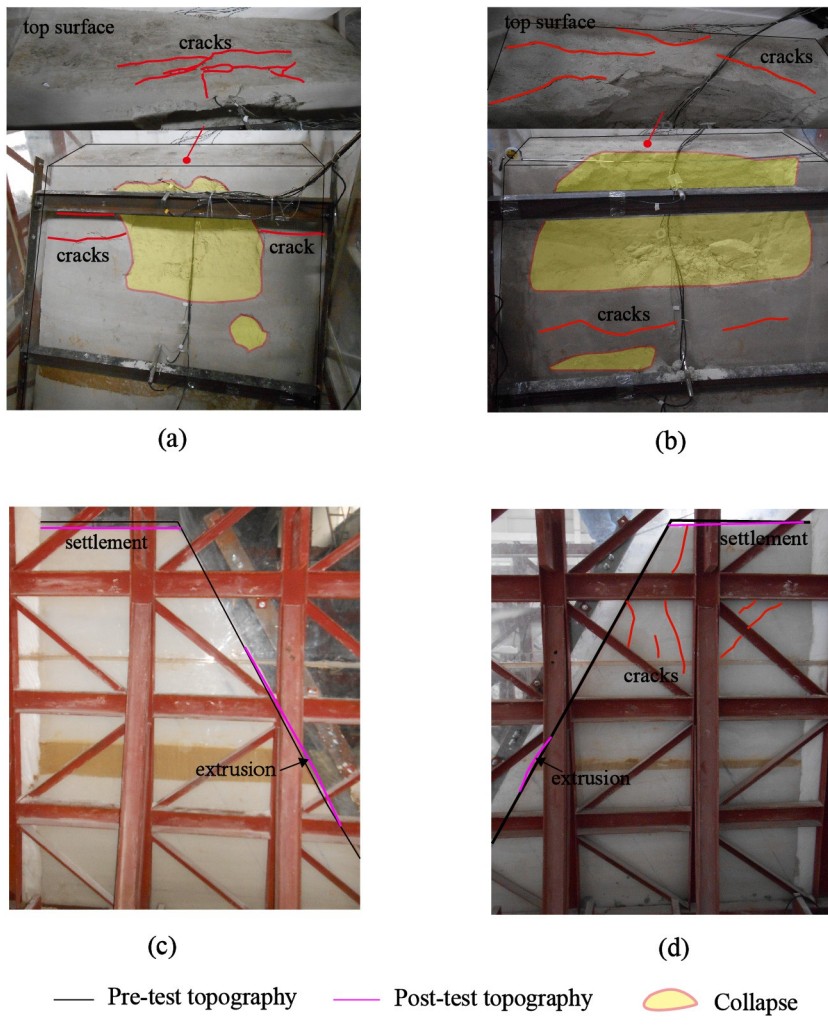

**Fig 15.** Final deformation and failure (Phase Ⅳ;) of two model slopes: (a) surface of Slope A; (b) surface of Slope B; (c) side view of Slope A; (d) side view of Slope B.

interlayer), highlighting the effect of the weak interlayer as limited seismic energy could propagate across the interlayer. Under vertical acceleration, the topographic amplification mainly occurred in the lower half of Slope A but in the upper half of Slope B.

(2) As the horizontal input amplitude increased, the maximum horizontal acceleration amplification occurred beneath the crest in Slope A but at the crest in Slope B. Both slopes exhibited a time lag of horizontal accelerations in the upper slope relative to the lower slope and an imbalance between inward and outward horizontal movements, suggesting a nonlinear response in the upper slope under strong horizontal shaking.

(3) Under all levels of vertical shaking, both slopes exhibited a maximum vertical acceleration amplification at the crest. The slope surface and slope interior experienced synchronous vertical acceleration response for Slope B whereas there was a minor time lag between slope surface and slope interior for Slope A. Nonlinear response was not significant in both slopes under vertical shaking.

(4) The dynamic response process of both slopes can be divided into four phases as the input acceleration increased. Phase I is the linear response phase, corresponding to input amplitudes lower than 0.3 g; Phase II is the transition phase from linear to nonlinear responses, corresponding to input amplitudes between 0.3 g and 0.5 g; Phase III is the nonlinear response phase, corresponding to input amplitudes larger than 0.5 g; and Phase IV is the final failure phase. Similar deformation characteristics including development of strong response zone and macro-cracks during Phase III, vertical settlement, horizontal extrusion and collapse in the upper half during Phase IV were observed in both slopes; however, the deformations were more severe in Slope B than in Slope A, suggesting an energy isolation effect of the thick interlayer in Slope A.

## Supporting information

**S1 Fig.** Fourier amplitude spectral ratios of horizontal accelerations (H-FASR) when input amplitude of horizontal shaking was lower than 0.3 g: (a) Slope A; (b) Slope B.
(TIF)

**S2 Fig. Horizontal displacement time histories at monitoring points D2′ and D2 under different levels of horizontal shaking.**
(TIF)

## Acknowledgments

Fei Zhou, Zheng Yang, Feng Wang, Hongjuan Hou and Xiaoqi Yang who assisted in the tests are thanked sincerely. The authors are grateful to two unnamed reviewers for their valuable comments on the early version of the manuscript.

## Author Contributions

**Conceptualization:** Hanxiang Liu.

**Data curation:** Hanxiang Liu.

**Methodology:** Qiang Xu.

**Writing – original draft:** Hanxiang Liu.

**Writing – review & editing:** Tong Qiu.

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
