## [Decision Letter · Decision Letter 0]

3 Dec 2020

PONE-D-20-32203

Dynamic acceleration response of a rock slope with a horizontal weak interlayer in shaking table tests

PLOS ONE

Dear Dr. liu,

Thank you for submitting your manuscript to PLOS ONE. After careful consideration, we feel that it has merit but does not fully meet PLOS ONE’s publication criteria as it currently stands. Therefore, we invite you to submit a revised version of the manuscript that addresses the points raised during the review process.

We look forward to receiving your revised manuscript.

Kind regards,

Jianguo Wang, PhD

Academic Editor

PLOS ONE

Journal Requirements:

2. Please include a caption for figure 3.

Reviewers' comments:

Reviewer's Responses to Questions

**Comments to the Author**

1. Is the manuscript technically sound, and do the data support the conclusions?

Reviewer #1: Yes

2. Has the statistical analysis been performed appropriately and rigorously? 

Reviewer #1: Yes

3. Have the authors made all data underlying the findings in their manuscript fully available?

Reviewer #1: Yes

4. Is the manuscript presented in an intelligible fashion and written in standard English?

Reviewer #1: Yes

5. Review Comments to the Author

Reviewer #1: In general, the topic of the paper is certainly of interest to the researchers of slope stability under earthquake. However, there are some unclear statements and questions that need to be clarified. The comments are listed below.

1.The resolution of the graphs provided in the article is not clear. It is recommended that the author provide graphs with high resolution and good quality.

2.Do the material parameters of the physical model listed in table 3 conform to the simulation similarity law listed in table 2?

3.A series of seismic loading tests are performed on the same model. How to ensure that the physical model will not affect the results of the next set of tests due to the deformation of the specimen, the generation of micro-cracks, and the interlayer sliding after the previous stage of the test?

4.In slope A, the thickness of the weak interlayer is 15 cm, E is 41 MPa (which is 1/10 of rock’s elastic modulus). Does the sliding between soft layer and rock or the deformation in the interlayer is observed during the first and second loading stages?

5.Figure 14 demonstrates the response features of the slope speculated according to the literature and this research. However, it is not clear how to the acceleration 0.3 g as a threshold value in Phase I.

6.There is a horizontal truss in Figure 15 and the crack range shown in Figure 15(a)(b) does not seem to be the failure type developed by the tension crack on the top of the slope. Does the truss collide with the physical model of the slope during the seismic loading?

6. PLOS authors have the option to publish the peer review history of their article (what does this mean?). If published, this will include your full peer review and any attached files.

Reviewer #1: No

---

## [Author Response · Author response to Decision Letter 0]

5 Jan 2021

Response to Review Comments

We have significantly revised our paper based on the constructive and helpful comments from the editors and reviewers. Our sincere thanks go to them for their suggestions and the opportunity for us to revise the manuscript. We have tried our best to address the review comments by making the following revisions. In the revised manuscript where the changes are tracked, all the revisions are highlighted in red.

Response to Editors

“1. Please include the following items when submitting your revised manuscript:

（1）A rebuttal letter that responds to each point raised by the academic editor and reviewer(s). You should upload this letter as a separate file labeled 'Response to Reviewers'.

（2）A marked-up copy of your manuscript that highlights changes made to the original version. You should upload this as a separate file labeled 'Revised Manuscript with Track Changes'.

（3）An unmarked version of your revised paper without tracked changes. You should upload this as a separate file labeled 'Manuscript'.”

Response: According to your instruction, three separate files have been uploaded: 'Response to Reviewers', 'Revised Manuscript with Track Changes', and 'Manuscript'. 

“2. ” 

Response: Thank you. No changes need to be made on the financial disclosure. 

“3. Guidelines for resubmitting your figure files are available below the reviewer comments at the end of this letter.”

Response: Thank you. All figures were resubmitted using the files adjusted by the Preflight Analysis and Conversion Engine (PACE) digital diagnostic tool.

“4. If applicable, we recommend that you deposit your laboratory protocols in protocols.io to enhance the reproducibility of your results. Protocols.io assigns your protocol its own identifier (DOI) so that it can be cited independently in the future. ”

Response: Thanks for your suggestion. No laboratory protocols were deposited in protocols.io and the detailed descriptions of test program can be found in the section 'Test program' in our revised paper. 

“5. Please ensure that your manuscript meets PLOS ONE's style requirements, including those for file naming.”

Response: The revised manuscript has been double-checked and submitted according to PLOS ONE’s submission guidelines and recently published papers in PLOS ONE. 

“5. Please include a caption for figure 3.”

Response: Thank you for catching our mistakes. A caption for Figure 3 has been added in the revised manuscript. Please see Page 9, Lines 222-224 in the revised manuscript without tracked changes.

“6. Please include captions for your Supporting Information files at the end of your manuscript, and update any in-text citations to match accordingly.”

Response: The supporting information files have been provided and cited in the revised paper according to the guidelines for Supporting Information in PLOS ONE. 

Response to Reviewer #1

“In general, the topic of the paper is certainly of interest to the researchers of slope stability under earthquake. However, there are some unclear statements and questions that need to be clarified. The comments are listed below.

1. The resolution of the graphs provided in the article is not clear. It is recommended that the author provide graphs with high resolution and good quality.”

Response: Thanks for your suggestion. All figures were resubmitted using the files adjusted by the Preflight Analysis and Conversion Engine (PACE) digital diagnostic tool, with high resolution and good quality.

“2. Do the material parameters of the physical model listed in table 3 conform to the simulation similarity law listed in table 2?” 

Response: Thanks for your comment. For the convenience of comparison, the ratios between prototype and model slopes for each parameter in Table 3 were calculated and shown in the following table. It can be seen that these ratios conformed to the corresponding scaling factors listed in Table 2, with an exception of cohesion, c. The cohesions of the rock and weak interlayer were much lower than the values required to satisfy the scaling factors of cohesion in Table 2. However, the dynamic behaviors of the model slopes, which are the focus of this present study, are not significantly influenced by the inconformity in cohesion, particularly under small deformations. 

We have added the statements in the section "Preparation of model slopes" to address this issue, please see Page 7, Lines 163-169 in the revised manuscript named as "Manuscript".

Table 3. Physical and mechanical properties of prototype slopes and model slopes.

Lithology Density ρ (103kg/m3) Cohesion c (kPa) Frictional angle ϕ (o) Elastic modulus E (MPa) Poisson’s ratio μ

Rock Prototype 2.48 520 33.8 1900.0 0.31

 Model 2.40 37.1 34.9 50.2 0.30

 Prototype/model 1.0 14.0 1.0 37.8 1.0 

 Scaling factor in Table 2 1.0 32.6 1.0 32.6 1.0

Weak interlayer Prototype 1.60 30.0 35.0 41.0 0.35

 Model 2.32 10.0 27.3 4.8 0.35

 Prototype/Model 0.7 3.0 1.3 8.5 1.0 

 Scaling factor in Table 2 0.67 10.67 1.0 10.67 1.0

“3. A series of seismic loading tests are performed on the same model. How to ensure that the physical model will not affect the results of the next set of tests due to the deformation of the specimen, the generation of micro-cracks, and the interlayer sliding after the previous stage of the test?”

Response: Thanks for your comment. You are right! It is widely known that loading history has a significant impact on the dynamic response of the model slopes. As a widely-adopted mitigation to reduce this impact, waves were applied to the model slopes with a gradually increasing amplitude [29-39]. The loading sequence in Table 4 allowed an insight into the evolution of the slope response from linear to nonlinear, assuming that the effect of minor deformation of model slope can be neglected under small-amplitude excitations. In addition, the cumulative damage caused by continuous loading could partially simulate pre-existing fractures/fissures in a prototype slope before a seismic event. 

We have added the statements in the section "Input excitations" to address this issue, please see Pages 8-9, Lines 205-212 in the revised paper named as "Manuscript".

“4. In slope A, the thickness of the weak interlayer is 15 cm, E is 41 MPa (which is 1/10 of rock’s elastic modulus). Does the sliding between soft layer and rock or the deformation in the interlayer is observed during the first and second loading stages?”

Response: Thanks for your comment. In the first stage, no sliding was observed between the weak interlayer and the surrounding rock, and no deformation was observed in the interlayer of both slopes. In the second loading stage, however, a small horizontal extrusion deformation occurred around the weak interlayer in both slopes, and the extrusion was more severe in Slope A (see Fig 15(C); S2 Fig) than in Slope B (see Fig 15(D); S2 Fig). The corresponding statements can be found in Page 18, Lines 469-472 in the revised paper named as "Manuscript".

“5. Figure 14 demonstrates the response features of the slope speculated according to the literature and this research. However, it is not clear how to the acceleration 0.3 g as a threshold value in Phase I.”

Response: Thanks for your comment. We have to acknowledge that the cutoff values of input acceleration between two successive response phases of the model slope can’t be accurately determined just based on our current research. Nonetheless, we selected acceleration 0.3 g as the threshold value in Phase I, considering the approximatively linear response characteristics of both slopes as the input amplitude was lower than 0.3 g:

(1) Both the horizontal and vertical accelerations (e.g., PHA, R-PHA, PVA, R-PVA) at most monitoring points of the slopes generally increased as the input amplitude increased from 0.1 g to 0.3g (see Fig 7 and Fig 11). 

(2) Minor change of resonant frequency occurred in both slopes as the input amplitude was increased from 0.1 g to 0.3g, suggesting a negligible structure deterioration of both slopes (see S1 Fig). 

(3) No macro deformation occurred in any parts of the slopes. 

The corresponding statements can be found in Page 17, Lines 437-444 in the revised paper named as "Manuscript".

“6. There is a horizontal truss in Figure 15 and the crack range shown in Figure 15(a)(b) does not seem to be the failure type developed by the tension crack on the top of the slope. Does the truss collide with the physical model of the slope during the seismic loading?”

Response: Thanks for your comment. The truss didn’t collide with the model slopes during the seismic loading due to a sufficient horizontal distance between the truss and the slope surface (see Fig A1 below). The distance is larger than the maximum horizontal displacement out of slope (see S2 Fig). 

Fig. A1. Horizontal distance between truss and slope surface.

S2 Fig. Horizontal displacement time histories at monitoring points D2′ and D2 under different levels of horizontal shaking.

---

## [Decision Letter · Decision Letter 1]

10 Feb 2021

PONE-D-20-32203R1

Dynamic acceleration response of a rock slope with a horizontal weak interlayer in shaking table tests

PLOS ONE

Dear Dr. liu,

Thank you for submitting your manuscript to PLOS ONE. After careful consideration, we feel that it has merit but does not fully meet PLOS ONE’s publication criteria as it currently stands. Therefore, we invite you to submit a revised version of the manuscript that addresses the points raised during the review process.

ACADEMIC EDITOR:

The second reviewer still has some comments to be addressed.

We look forward to receiving your revised manuscript.

Kind regards,

Jianguo Wang, PhD

Academic Editor

PLOS ONE

Reviewers' comments:

Reviewer's Responses to Questions

**Comments to the Author**

1. If the authors have adequately addressed your comments raised in a previous round of review and you feel that this manuscript is now acceptable for publication, you may indicate that here to bypass the “Comments to the Author” section, enter your conflict of interest statement in the “Confidential to Editor” section, and submit your "Accept" recommendation.

Reviewer #1: All comments have been addressed

Reviewer #2: All comments have been addressed

2. Is the manuscript technically sound, and do the data support the conclusions?

Reviewer #1: Yes

Reviewer #2: Yes

3. Has the statistical analysis been performed appropriately and rigorously? 

Reviewer #1: Yes

Reviewer #2: Yes

4. Have the authors made all data underlying the findings in their manuscript fully available?

Reviewer #1: Yes

Reviewer #2: Yes

5. Is the manuscript presented in an intelligible fashion and written in standard English?

Reviewer #1: Yes

Reviewer #2: Yes

6. Review Comments to the Author

Reviewer #1: (No Response)

Reviewer #2: (No Response)

7. PLOS authors have the option to publish the peer review history of their article (what does this mean?). If published, this will include your full peer review and any attached files.

Reviewer #1: No

Reviewer #2: No

---

## [Author Response · Author response to Decision Letter 1]

28 Mar 2021

Response to Review Comments

We have significantly revised our paper based on the constructive and helpful comments from the editors and reviewers. Our sincere thanks go to them for their suggestions and the opportunity for us to revise the manuscript. We have tried our best to address them by making the following revisions. In the revised paper, all the revisions are highlighted in red.

Response to Editors

“1. Please include the following items when submitting your revised manuscript:

A rebuttal letter that responds to each point raised by the academic editor and reviewer(s). You should upload this letter as a separate file labeled 'Response to Reviewers'.

A marked-up copy of your manuscript that highlights changes made to the original version. You should upload this as a separate file labeled 'Revised Manuscript with Track Changes'.

An unmarked version of your revised paper without tracked changes. You should upload this as a separate file labeled 'Manuscript'.”

Response: According to the guidelines for revisions, three separate files have been uploaded named as 'Response to Reviewers', 'Revised Manuscript with Track Changes', and 'Manuscript', respectively. 

“2. ” 

Response: Thank you. No changes need to be made on the financial disclosure. 

“3. Guidelines for resubmitting your figure files are available below the reviewer comments at the end of this letter.”

Response: Thank you. All figures were submitted using the files adjusted by the Preflight Analysis and Conversion Engine (PACE) digital diagnostic tool.

“4. If applicable, we recommend that you deposit your laboratory protocols in protocols.io to enhance the reproducibility of your results. Protocols.io assigns your protocol its own identifier (DOI) so that it can be cited independently in the future. ”

Response: Thanks for your suggestion. No laboratory protocols were deposited in protocols.io and the detailed descriptions of test program can be found in Caption 'Test program' in our revised paper. 

Response to Reviewer 

“The dynamic response of a slope having a weak interlayer is an interesting but complex research topic in the slope earthquake engineering field. The research work in this paper can help understand the failure mechanism of geohazards with this type of slope structure under an earthquake. The model test program was designed rigorously and the results of model tests were analyzed in detail. The language is fluent and easy to understand, but there are still some aspects that need to be improved. My general comments are as follows:

 (1) The shaking table test was conducted for a limited slope model geometry (regular single-side slope) and slope structure (a single horizontal interlayer). The authors are encouraged to carry out further research under more complex conditions (e.g., complex topography, slope structure, and seismic loading) so that the results obtained can be generalizable.”

Response: Thanks for your suggestion. You are right! The seismic slope responses are influenced by diverse factors such as topography, slope structure, and seismic loading. In the present study, we simplified the slope model geometry and the slope structure to highlight the effect of a weak interlayer on the complex seismic response and deformation/failure behavior of rock slopes. In the future work, an extensive numerical calculation will be carried out to study the seismic slope responses under more complex conditions using the numerical slope models calibrated by these shaking table test results, together with the parametric analysis. The reason for this simplification can be found in Page 4, Lines 97-106 in the revised paper named as "Manuscript".

“(2) As shown in Table 4, why are the maximum input amplitudes of vertical and horizontal accelerations different?” 

Response: Thanks for your comment. The target input amplitudes of vertical accelerations were designed to be the same with those of horizontal accelerations. However, the actual input amplitudes of vertical accelerations within the effective working frequency (< 80 Hz), which were collected by an accelerometer fixed on the shaking table, were lower than the target input amplitudes due to the capacity of the shaking table under high levels of vertical excitations. The target and actual input amplitudes of the vertical accelerations are shown in Table A1. The corresponding statement can be found in Page 9, Lines 224-228 in the revised paper named as "Manuscript".

Table A1. Targeted and actual input amplitudes of vertical accelerations in shaking table tests (revised from Table 4)

loading step Wave type Target input amplitude (g) Actual input amplitude (g) Excitation direction Duration (s) Dominant frequency (Hz)

1 WL wave 0.1 0.1 Z 27 32.4

3 WL wave 0.2 0.18 Z 27 32.4

5 WL wave 0.3 0.24 Z 27 32.4

7 WL wave 0.4 0.32 Z 27 32.4

9 WL wave 0.5 0.38 Z 27 32.4

11 WL wave 0.8 0.6 Z 27 32.4

13 WL wave 1.0 0.75 Z 27 32.4

“(3) Since the seismic input was from the Wolong station – the authors should explain the geological/geomorphological setting: is on the surface? on bedrock, what type? on flat/valley bottom area?”

Response: Thanks for your comment. We have added the following statement in the revised paper to describe the basic characteristics of the Wolong station:

Page 9, Lines 217-224: The input accelerations were scaled from the recorded accelerations at the Wolong seismic station (WL wave) during the 2008 Wenchuan earthquake, which was about 23 km southwest of the epicenter. The altitude on the station ground is 919 m above sea level whereas the altitude on the top of the mountain nearby is 3187 m. The overlying soils at the station are mainly composed of Quaternary alluvial and diluvial gravels and pebbles. The recorded accelerations had a dominant frequency of 2.4 Hz and 8.1 Hz in the horizontal and vertical direction, respectively, which were scaled to have dominant frequency of 9.6 Hz and 32.4 Hz, respectively (see Table 4).

“(4) Lastly, the present work focused on the acceleration responses of the slopes. It would have been nice to see some quantitative work on the failure mechanisms and failure modes of the slopes.”

Response: Thanks for your comment. We have to acknowledge that a completely quantitative analysis was not carried out with respect to the failure mechanisms and failure modes of the slopes, just based on the macroscopic observations in a shaking table test. The quantitative work on the seismic slope responses and failure modes and mechanisms will be carried out in the future using different research methods, including macro- and micro- numerical calculation, physical model testing, field monitoring, and theoretical analysis. We have added the corresponding statement in Pages 18-19, Lines 487-491 in the revised paper named as "Manuscript".

---

## [Decision Letter · Decision Letter 2]

7 Apr 2021

Dynamic acceleration response of a rock slope with a horizontal weak interlayer in shaking table tests

PONE-D-20-32203R2

Dear Dr. liu,

We’re pleased to inform you that your manuscript has been judged scientifically suitable for publication and will be formally accepted for publication once it meets all outstanding technical requirements.

Kind regards,

Jianguo Wang, PhD

Academic Editor

PLOS ONE

Additional Editor Comments (optional):

Reviewers' comments:

Reviewer's Responses to Questions

**Comments to the Author**

1. If the authors have adequately addressed your comments raised in a previous round of review and you feel that this manuscript is now acceptable for publication, you may indicate that here to bypass the “Comments to the Author” section, enter your conflict of interest statement in the “Confidential to Editor” section, and submit your "Accept" recommendation.

Reviewer #2: All comments have been addressed

2. Is the manuscript technically sound, and do the data support the conclusions?

Reviewer #2: Yes

3. Has the statistical analysis been performed appropriately and rigorously? 

Reviewer #2: Yes

4. Have the authors made all data underlying the findings in their manuscript fully available?

Reviewer #2: Yes

5. Is the manuscript presented in an intelligible fashion and written in standard English?

Reviewer #2: Yes

6. Review Comments to the Author

Reviewer #2: (No Response)

7. PLOS authors have the option to publish the peer review history of their article (what does this mean?). If published, this will include your full peer review and any attached files.

Reviewer #2: No

---

## [Editor Report · Acceptance letter]

12 Apr 2021

PONE-D-20-32203R2 

Dynamic acceleration response of a rock slope with a horizontal weak interlayer in shaking table tests 

Dear Dr. liu:

I'm pleased to inform you that your manuscript has been deemed suitable for publication in PLOS ONE. Congratulations! Your manuscript is now with our production department. 

Kind regards, 

on behalf of

Dr. Jianguo Wang 

Academic Editor

PLOS ONE